# A brainstem to circadian system circuit links Tau pathology to sundowning-related disturbances in an Alzheimer's disease mouse model

Andrew E. Warfield[1], Pooja Gupta[1], Madison M. Ruhmann[1], Quiana L. Jeffs[1], Genevieve C. Guidone[1], Hannah W. Rhymes[1], McKenzi I. Thompson[2] & William D. Todd ®[1] ✉

Alzheimer's disease (AD) patients exhibit progressive disruption of entrained circadian rhythms and an aberrant circadian input pathway may underlie such dysfunction. Here we examine AD-related pathology and circadian dysfunction in the APPSwe-Tau (TAPP) model of AD. We show these mice exhibit phase delayed body temperature and locomotor activity with increases around the active-to-rest phase transition. Similar AD-related disruptions are associated with sundowning, characterized by late afternoon and early evening agitation and aggression, and we show TAPP mice exhibit increased aggression around this transition. We show such circadian dysfunction and aggression coincide with hyperphosphorylated Tau (pTau) development in lateral parabrachial (LPB) neurons, with these disturbances appearing earlier in females. Finally, we show LPB neurons, including those expressing dynorphin (LPB$^{dyn}$), project to circadian structures and are affected by pTau, and LPB$^{dyn}$ ablations partially recapitulate the hyperthermia of TAPP mice. Altogether we link pTau in a brainstem circadian input pathway to AD-related disturbances relevant to sundowning.

Alzheimer's disease (AD) affects between 10 and 30% of the population over the age of 65 and treatments for AD are mostly limited to symptom management[1]. The estimated annual cost of AD in 2022 is 321 billion dollars and this is projected to increase to almost 1 trillion dollars annually by 2050. Much of this cost is due to the need for institutionalization, and the average annual per-person payment of a patient in a nursing home is about 25 times higher for people with AD or dementia compared to those without[2]. Chief risk factors for institutionalization include sleep and circadian disorders[3], which have been seen clinically in AD patients for decades[4], and recently have been theorized as potentially causative for AD pathogenesis[1]. In fact, circadian and sleep disturbances have been shown to predict the onset of dementia and cognitive impairment[4,5], and worsen throughout the disease course. One of the most common circadian-related changes in symptomatic AD patients is a phase delay, revealed by later bathyphases (troughs) and acrophases (peaks) of entrained rhythms including body temperature (Tb) and locomotor activity (LMA)[6–8]. This phase delay is also a strong predictor of sundowning, a poorly understood circadian syndrome seen in 20–25% of AD patients[9,10]. Sundowning is most commonly characterized by agitation, behavioral aggression, and increased LMA in the form of wandering during the late afternoon and early evening.

Circadian rhythms in mammals are mediated by the master pacemaker, the suprachiasmatic nucleus (SCN) of the hypothalamus,

[1]Department of Zoology and Physiology, Program in Neuroscience, University of Wyoming, Laramie, WY, USA. [2]Department of Epidemiology, Rollins School of Public Health, Emory University, Atlanta, GA, USA. ✉e-mail: wtodd3@uwyo.edu

which is entrained to the daily light-dark cycle via direct retinal projections[11]. The SCN synchronizes downstream structures mainly through its obligate axonal relay in the adjacent subparaventricular zone (SPZ)[12–14]. Together, the SCN and SPZ regulate rhythms of Tb, LMA, sleep-wake, feeding, corticosterone release, and even the propensity for behavioral aggression[12,15,16]. Healthy aging in humans is also associated with disruption of entrained circadian rhythms. In work combining actigraphy and analyses of postmortem hypothalamic tissue, such age-related circadian disruption strongly correlates with a loss of SCN neurons that express vasoactive intestinal peptide (SCN[VIP])[7]—which have been shown in mouse studies to play a primary role as circadian pacemakers[17]. While AD patients exhibit the expected phase delay, they do not show an additional loss of SCN[VIP] neurons[7,18]. This suggests that an input to the circadian system may instead be the site of the AD-related pathology and neurodegeneration that underlies AD-related circadian dysfunction, and potentially sundowning. Several areas known to send inputs to the circadian system have been shown to be affected by AD pathology in both patients and mouse models[10]. However, to our knowledge, no study has combined analyses of the onset of circadian dysfunction and AD-pathology in conjunction with circuit-based mapping of circadian inputs in the same model.

Here we examine the onset of circadian dysfunction in an AD mouse model that expresses the human 695-amino acid isoform of the amyloid precursor protein (APP) as well as the human P301L mutation of the microtubule-associated protein tau (MAPT) gene (TAPP mice, also known as APPSwe-Tau, Taconic Model 2469)[19]. TAPP mice thus develop both extracellular plaques consisting of amyloid-beta (Aβ) and intracellular neurofibrillary tangles consisting of hyperphosphorylated Tau (pTau), the two pathological hallmarks of AD. TAPP mice have also been shown to exhibit cognitive impairment[20]. We particularly focus on sex differences in the onset of circadian dysfunction and AD pathology in this model, as there are marked sex differences described in female versus male AD patients[9,21,22]. In the general population, the prevalence of insomnia and circadian sleep/wake issues is greater in women than it is in men[23]. Additionally, two-thirds of AD patients are women, and cognitive decline is often worse in women with AD than in men with AD[21]. Furthermore, women with AD often exhibit more neuropsychiatric behavioral disruptions than men with AD[22], and sundowning symptoms are also more prevalently reported in women[9].

We find that TAPP mice develop a phase delay in entrained Tb and LMA rhythms, similar to that reported in AD patients[6,7,24], and that TAPP females express this circadian dysfunction earlier than males. This phase delay is also characterized by increases in Tb and LMA around the dark-to-light transition (the active-to-rest phase transition in nocturnal mice), which is temporally analogous to when AD patients show sundowning symptoms. We also show that both TAPP females and males show increased behavioral aggression around this transition at ages when this phase delay is present. Importantly, these circadian disruptions in TAPP mice coincide with the development of intracellular pTau pathology and extracellular plaques expressing Aβ. Using retrograde tracing from the SCN and SPZ we determined which inputs to the circadian system also showed AD pathology in conjunction with circadian dysfunction. The area we identified that most consistently displayed AD pathology and also projected to the circadian system was the lateral parabrachial nucleus (LPB) of the brainstem. Notably, we found that the LPB exhibited pTau pathology in all TAPP mice that showed circadian dysfunction, and both circadian dysfunction and LPB pTau develop up to four months earlier in TAPP females. Using genetically targeted Cre-dependent anterograde tracing, and RNAscope in situ hybridization combined with retrograde tracing, we further show that the dynorphin-expressing subpopulation of LPB (LPB[dyn]) neurons strongly projects to the structures of the circadian system. Finally, we demonstrate that these LPB[dyn] neurons are affected by pTau pathology in TAPP mice and that they show

neurodegeneration in some TAPP mice at later ages. Altogether, here we characterize a circadian input pathway that is affected by pTau pathology which coincides with the development of circadian physiological and behavioral disturbances in an age- and sex-dependent manner.

## Results

### TAPP mice exhibit a phase delay in Tb and LMA as well as marked changes in overall rhythms when Tau pathology is present in the LPB

One of the most common circadian phenotypes reported in AD patients is a phase delay, characterized by later bathyphases (daily minimums/troughs) of Tb and later acrophases (daily maximums/peaks) of LMA[6–8]. Given the relative paucity of AD models that recapitulate this phenotype[8], we sought to investigate whether TAPP mice showed phase delays of Tb and LMA compared to WT controls. Furthermore, we investigated the relationship of such circadian function with the presence or absence of AD-related pathology (pTau and Aβ) throughout the brain.

In order to observe the time-dependent effects of AD-related pathology on daily Tb and LMA rhythms, we analyzed daily hr-by-hr data averaged over 7 days in LD and overlaid these curves in both 3–5 month-old (3–5mo) and 7–9 month-old (7–9mo) TAPP and WT mice. We also analyzed Tb bathyphase and LMA acrophase as markers of circadian phase because these are among the most commonly used phase markers in AD patients[6] (Fig. 1). We saw that hr-by-hr values were not significantly different between 3–5mo male TAPP and WT mice (Fig. 1a, c). We also found that Tb bathyphase in 3–5mo males was not statistically different between TAPP and WT mice, however LMA acrophase was slightly, but significantly, later in TAPP males at this age (Fig. 1b, d). In 7–9mo TAPP males we saw increases in Tb and LMA at multiple time points throughout the dark/active phase, and this hyperthermia and hyperactivity continued into the early light/rest phase (Fig. 1e, g). This also corresponded with a significant phase delay in both Tb bathyphase and LMA acrophase of male 7–9mo TAPP mice compared to male WT controls (Fig. 1f, h). We saw similar age-dependent differences in Tb acrophase and LMA bathyphase as well (Table S1).

To our surprise, in TAPP females we saw circadian dysfunction in both the 3–5mo and 7–9mo groups. 3–5mo TAPP females showed a decrease in Tb at ZT10 and an increase at ZT23, indicative of a rightward shifting of the Tb rhythm (phase delay). For LMA we saw a decrease around the rest-to-active phase transition and a trend towards increased activity at the active-to-rest phase transition, similarly indicating a rightward shift of the LMA curve (phase delay, Fig. 1i, k). In the 7–9mo TAPP females these changes were amplified. These older TAPP females experienced hyperthermia during the two hours before and after the active-to-rest phase transition, as well as an increase in LMA at multiple time points throughout the dark phase (Fig. 1m, o). We also saw a phase delay in Tb and LMA at 3–5mo (Fig. 1j, l) and in Tb at 7–9mo though this was not significant for LMA at this age due to later acrophases in the WT females (Fig. 1n, p, Fig. S1k). As expected, we saw similar results when we examined Tb acrophase and LMA bathyphase (Table S1). Altogether these data indicated that TAPP mice exhibit a phase delay in both Tb and LMA, symptoms commonly associated with sundowning in AD, in a sex and age dependent manner.

We further sought to evaluate these sex and age effects directly (Fig. S1). In making direct comparisons between males and females we found that 3–5mo TAPP females had later Tb bathyphases compared to 3–5mo TAPP males (Fig. S1a). Mice in the 7–9mo groups were not different from each other in either Tb bathyphase or LMA acrophase (Fig. S1e–h). These comparisons further support the phase delays shown in Fig. 1. When we compared mice by age we found that 7–9mo WT females were delayed compared to 3–5mo WT females, likely due

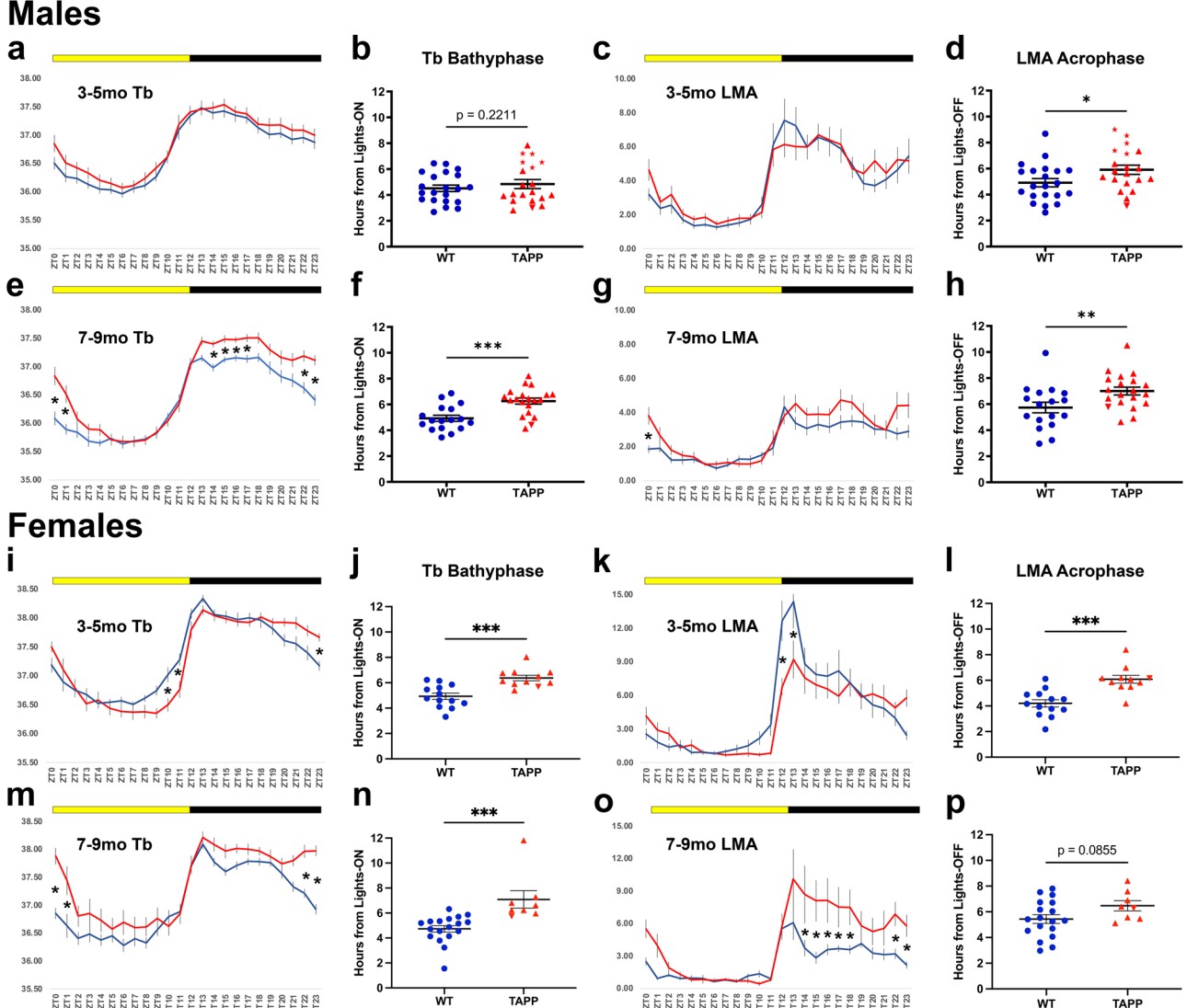

**Fig. 1 | Female TAPP mice exhibit phase delay earlier than males. a, b** 3–5mo TAPP males (red, $n = 20$) showed no hr-by-hr body temperature (Tb) differences compared to wildtypes (WT, blue, $n = 21$) [Two-way RM ANOVA, Interaction: $F_{(23,897)} = 0.3043$, $p = 0.9994$], nor later bathyphases [troughs; hours from lights-on, zeitgeber time (ZT)0] [Two-tailed unpaired $t$ test, $t_{(39)} = 0.7763$]. **c, d** 3–5mo TAPP males (red, $n = 20$) showed no hr-by-hr locomotor activity (LMA) count differences compared to WTs (blue, $n = 21$) [Two-way RM ANOVA, Interaction: $F_{(23,897)} = 0.7714$, $p = 0.7696$], but later acrophases (peaks; hours from lights-off, ZT12) [Two-tailed unpaired $t$ test, $t_{(39)} = 2.097$, *$p = 0.0213$], driven by a relevant subset (red stars in **b, d**; see also Figs. 2,4). **e, f** 7–9mo TAPP males (red, $n = 20$) showed increased Tb during the dark-to-light transition and early active phase compared to WTs (blue, $n = 17$) [Two-way RM ANOVA, Interaction: $F_{(23,805)} = 4.170$, $p = 0.0001$, Sidak's *post hoc*: $^{ZT0}p = 0.0198$, $^{ZT1}p = 0399$, $^{ZT14-ZT15}p = 0.0013$; $^{ZT16}p = 0.0052$, $^{ZT17}p = 0.0196$; $^{ZT22}p = 0.0111$, $^{ZT23}p = 0.0002$], and later bathyphases [Two-tailed unpaired $t$ test, $t_{(35)} = 3.983$, ***$p = 0.0002$]. **g, h** 7–9mo TAPP males (red, $n = 20$) showed increased ZT0 LMA compared to WTs (blue, $n = 17$) [Two-way RM ANOVA, Interaction: $F_{(23,805)} = 1.711$, $p = 0.0203$, Sidak's *post hoc*: $p = 0.0465$], and later acrophases [Two-tailed unpaired $t$ tests, $t_{(35)} = 2.528$, **$p = 0.0081$]. **i,j** 3–5mo TAPP females (red,

$n = 11$) showed increased ZT23 Tb and decreased ZT10-ZT11 Tb compared to WTs (blue, $n = 13$) [Two-way RM ANOVA, Interaction: $F_{(23,506)} = 4.535$, $p = 0.0001$, Sidak's *post hoc*: $^{ZT10}p = 0.0167$, $^{ZT11}p = 0.0202$, $^{ZT23}p = 0.0355$], and later bathyphases [Two-tailed unpaired $t$ test, $t_{(22)} = 4.295$, ***$p = 0.0003$]. **k, l** 3–5mo TAPP females (red, $n = 11$) showed decreased ZT12-ZT13 LMA compared to WTs (blue, $n = 13$) with a trend towards increased ZT23 LMA [Two-way RM ANOVA, Interaction: $F_{(23,506)} = 3.099$, $p = 0.0001$, Sidak's *post hoc*: $^{ZT12}p = 0.0003$; $^{ZT13}p = 0.0031$], and later acrophases [two-tailed unpaired $t$ test, $t_{(22)} = 4.455$, ***$p = 0.0002$]. **m, n** 7–9mo TAPP females (red, $n = 8$) showed increased ZT22-ZT1 Tb compared to WTs (blue, $n = 18$) [two-way RM ANOVA, Interaction: $F_{(23,552)} = 5.683$, $p < 0.0001$, *post hoc*: $^{ZT22}p = 0.0002$, $^{ZT23-ZT1}p < 0.0001$], and later bathyphases [two-tailed unpaired $t$ test, $t_{(24)} = 3.911$, ***$p = 0.0007$]. **o, p** 7–9mo TAPP females (red, $n = 8$) showed increased ZT22-ZT23 and ZT13-ZT18 LMA compared to WTs (blue, $n = 18$) [two-way RM ANOVA, Interaction: $F_{(23,552)} = 4.715$, $p < 0.0001$, Sidak's *post hoc*: $^{ZT13}p = 0.0005$, $^{ZT14-ZT15}p < 0.0001$, $^{ZT16}p = 0.0003$, $^{ZT17}p = 0.0178$, $^{ZT18}p = 0.0054$, $^{ZT22}p = 0.0096$, $^{ZT23}p = 0.0258$], but not later acrophases [Two-tailed unpaired $t$ test, $t_{(24)} = 1.793$]. Data are means ± SEM. Source data are provided as a Source Data file.

to natural aging (Fig. S1k). This supports a potential reason for why LMA acrophase was not significantly delayed between 7–9mo TAPP females compared to 7–9mo WT females in Fig. 1p. 7–9mo TAPP males, on the other hand, were delayed compared to 3–5mo TAPP males in both Tb bathyphase and LMA acrophase (Fig. S1n, p). Male WT mice did not differ at either age (Fig. S1m, o). In order to analyze the stability of Tb bathyphase and LMA acrophase we also compared standard

deviations in Tb bathyphase and LMA acrophase across the seven days of recording (Fig. S2). The only difference we found was that the 3–5mo TAPP males had more variable Tb bathyphases than WT controls (Fig. S2e).

We next sought to investigate what underlying pathology could be leading to these phase delays. To do this we stained the brains of these mice with antibodies for both pTau and Aß and compared the

locations of this revealed pathology to areas of the brain that are known to be affected by AD pathology early on in humans as well as in structures known to project to the circadian system. We found consistent expression of pTau in the LPB, an area that is affected by pTau pathology in preclinical stages and throughout the disease course in AD and ultimately is subject to neurodegeneration[25–27]. Importantly, the appearance of pTau in the LPB of TAPP mice is tightly correlated with the appearance of the phase delay and changes to our hr-by-hr graphs (Figs. 2a, b, S3). Five of the twenty 3–5mo TAPP males tested had already developed pTau in the LPB by time of recording (Fig. 2b, blue stars), while we did not detect consistent pTau anywhere in the brain in the other 15 at this age. The less consistent pTau accumulation in the LPB in 3–5mo TAPP males corresponds to the lack of an overall phase delay in Tb bathyphase compared to WT mice (Fig. 1b). However, LMA acrophase was significantly later in 3–5mo TAPP males compared to WT males, but this appeared to be driven by the 5 TAPP males that already exhibited LPB pTau (see red stars in Fig. 1d). LPB pTau accumulation was consistently present, in conjunction with the observed phase delays in Fig. 1, at each of the other age and sex combinations (Fig. 2b): 7–9mo male TAPP (blue squares), 3–5mo female TAPP (pink triangles), and 7–9mo female TAPP (pink squares). To better understand how Tb bathyphase or LMA acrophase change with levels of pTau accumulation in the LPB, we correlated these two phase markers with the average area covered by pTau in the LPB (Fig. 2b). We found a positive relationship between the log of either phase marker of circadian entrainment (Tb bathyphase, or LMA acrophase) and the log of area covered in the LPB by pTau (data in Fig. 2b are displayed in natural units). In addition to earlier LPB pTau accumulation, TAPP females also showed higher attrition rates than males (30% compared to 5%) and most of the TAPP females that died before perfusion were in the 3–5mo group.

Given the observed phase delays that correlated strongly with the appearance and amount of pTau in the LPB, we looked at individual actograms to better visualize these delays (Figs. 3, S4). Here it is evident, at the age and sex combinations where pTau is present in the LPB, that there is a rightward shift of Tb bathyphase (Fig. 3) and LMA acrophase (Fig. S4) compared to WT controls. We also analyzed Tb and LMA rhythms under free running conditions for 10d after at least 2 weeks in DD (black vertical bars in Figs. 3, S3). Interestingly, we did not find any significant differences in period length between TAPP and WT mice at either age or in either sex (Table S2). This may indicate that the circadian dysfunction seen in TAPP mice, and in AD patients for whom testing under DD conditions is often impractical, reflects primarily a disruption of entrained circadian rhythms.

To further investigate the differences between mice expressing pathology vs those without, we more closely examined the subset ($n = 5$) of 3–5mo TAPP males that exhibited pTau in the LPB. When we compared this subset to 3–5mo WT males, we saw that both Tb bathyphase and LMA acrophase were significantly delayed in 3–5mo TAPP males with LPB pTau (Fig. 4a, b). We also saw stark differences in hr-by-hr graphs with this subset of 3–5mo TAPP males displaying a clear rightward shifting of the curve, again indicative of a phase delay. Additionally, this phase delay was associated with hyperthermia at the hours around the active-to-rest phase transition and a decrease in both Tb and LMA counts at the hours around the rest-to-active phase transition (Fig. 4c, d). When compared to 3–5mo TAPP males who had not yet developed pTau in the LPB at the time of staining, we saw the same pattern as when compared to 3–5mo WT males. The 3–5mo TAPP males with LPB pTau continued to show significantly delayed Tb bathyphase and LMA acrophase (Fig. 4e, f). Tb was also increased at the hours around the active-to-rest phase transition, and both Tb and LMA were decreased at the hours around the rest-to-active phase transition

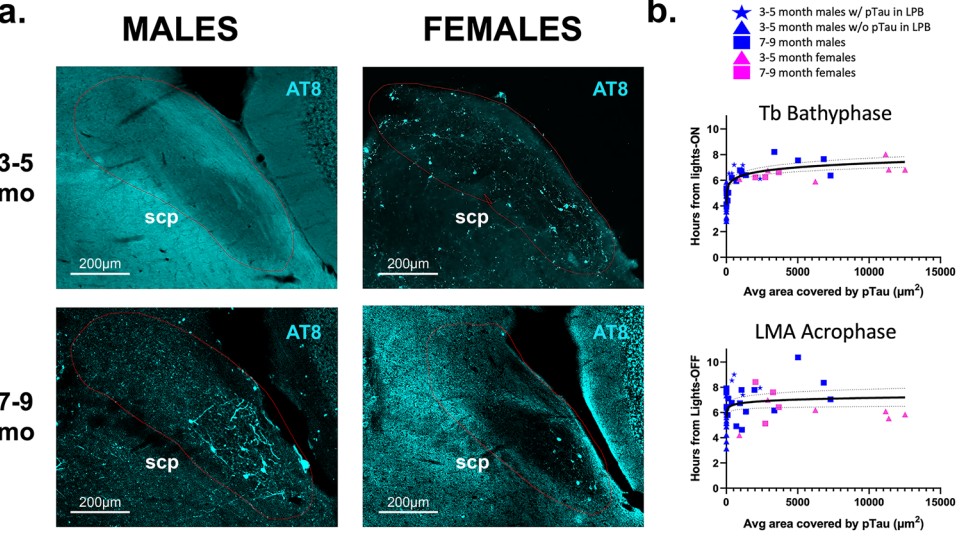

**Fig. 2 | The presence of pTau in the LPB is associated with the appearance of sundowning-like phase delays in body temperature and locomotor activity.**
**a** Male TAPP mice do not consistently exhibit pTau pathology (revealed by immunohistochemistry using the AT8 antibody, turquoise) in the LPB at 3–5 months (top left, $n = 15$ mice) but do at 7–9 months (bottom left, $n = 16$ mice). However, female TAPP mice exhibit pTau pathology in the LPB as early as 3–5 months (top right, $n = 6$ mice). Females continue to exhibit such pTau after 7–9 months (bottom right, $n = 4$ mice) but this appears to be reduced, perhaps due to neurodegeneration. Boundaries of the LPB are lightly drawn in red. Examples of the quantification of these same images is presented in Fig. S3. **b** Regression correlating the average area covered by pTau staining in the LPB with either Tb bathyphase values (top) or LMA acrophase values (bottom). TAPP females at 3–5mo (pink triangles, $n = 6$ mice) had the greatest area covered by pTau, followed by TAPP males at 7–9mo (blue squares, $n = 16$ mice). The decrease in area covered by

pTau in 7–9mo TAPP females (pink squares, $n = 4$ mice) is perhaps due to cell death associated with prolonged pTau expression. Most TAPP males at 3–5mo had no detectable pTau (blue triangles, $n = 15$ mice). However, a subset of 3–5mo TAPP males that did exhibit small levels of pTau (blue stars, $n = 5$ mice) were noticeably more phase delayed compared to those without pTau at this age. This logarithmic relation may suggest that the phase delay could be a method of early detection of increasing levels of pTau in the LPB. Linear relationship is seen when both predictor and response are log transformed, however the data is displayed in natural units. [log-log regression with F test: $F_{(1,41)} = 131.3$ (Tb bathyphase) and 7.613 (LMA acrophase), $\beta_1 = 0.06337$ (Tb bathyphase) and 0.02659 (LMA acrophase), $p < 0.0001$ (Tb bathyphase) and $p = 0.0046$ (LMA acrophase), $R^2 = 0.7620$ (Tb bathyphase) and 0.1719 (LMA acrophase)]. Data are presented as means ± 95% CI. Source data are provided as a Source Data file. scp, superior cerebellar peduncle.

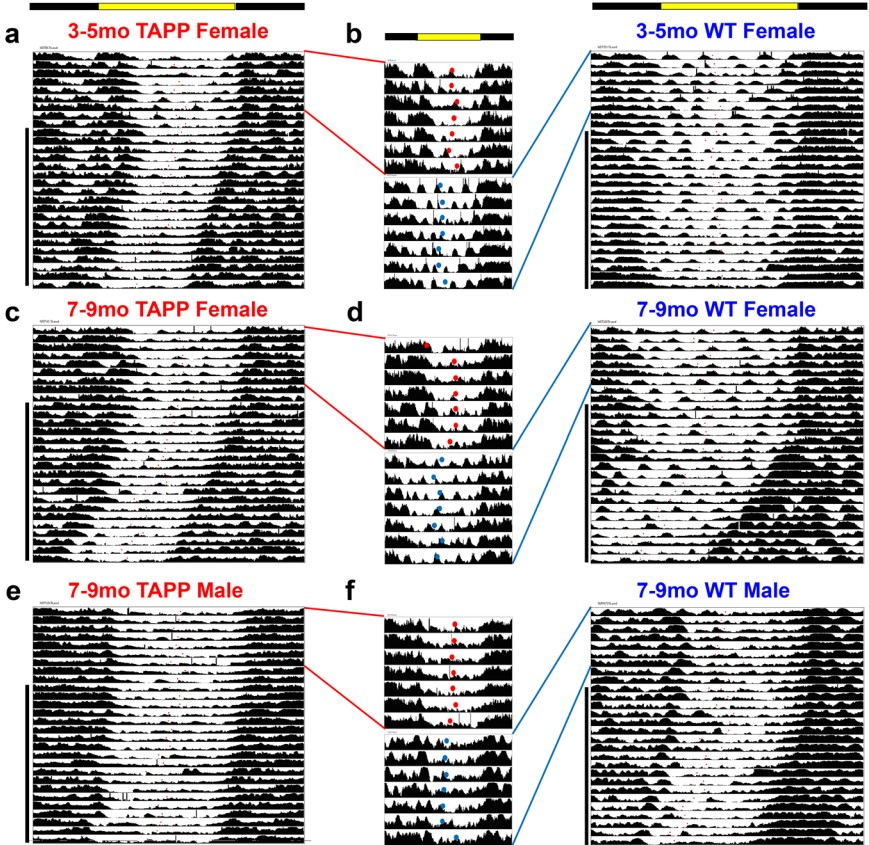

**Fig. 3 | Representative Tb actograms of 3–5mo and 7–9mo female TAPP vs WT mice and 7–9mo TAPP vs WT mice. a** Body temperature (Tb) recordings under 12 h–12 h light-dark (LD) and constant darkness (DD) of a 3–5mo TAPP female (left red, $n = 11$ mice) and 3–5mo WT female (right blue, $n = 13$ mice). **b** 7d of LD Tb recording depicting daily Tb bathyphase (dots) for a 3–5mo TAPP female (top red, $n = 13$ mice) and a 3–5mo WT female (bottom blue, $n = 11$ mice). **c** LD and DD recording of 7–9mo TAPP female (left red, $n = 8$ mice) and 7–9mo WT female (right blue, $n = 18$ mice). **d** 7d of LD Tb recording depicting daily Tb bathyphase (dots) for a 7–9mo TAPP female (top red, $n = 8$ mice) and 7–9mo WT female (bottom blue, $n = 18$ mice). **e** LD and DD recording of 7–9mo TAPP male (left red, $n = 20$ mice) and 7–9mo WT male (right blue, $n = 17$ mice). **f** 7d of LD Tb recording depicting daily Tb bathyphase (dots) for a 7–9mo TAPP male (top red, $n = 20$ mice) and 7–9mo WT male (bottom blue, $n = 17$ mice). 3–5mo males are shown in Fig. 4. Analysis of DD data is reported in Table S2.

(Fig. 4g, h). The apparent decrease in LMA around the resting-active phase transition in these mice could be due to the phase delay itself wherein this delay does not allow for mice to reach the peak of their LMA which often happens very early in the active phase. Finally, the subset of 3–5mo TAPP males with LPB pTau had differing levels of pathology (Fig. 4i), perhaps indicating that such circadian disturbances likely arise due to pTau-mediated alteration of overall firing dynamics within the LPB. When looking at individual actograms the phase delay displayed by mice with LPB pTau is evident compared to either WT controls or 3–5mo TAPP males without LPB pTau (Figs. 4j–l, S5). When we investigated the stability of Tb bathyphase and LMA acrophase in these five 3–5mo TAPP mice compared to either TAPP mice without LPB pTau or WT controls, we found that these mice were also more variable in their phase markers (Fig. S6). Together these data suggest two exciting ideas. First, the striking changes in entrained rhythms which are similarly seen in AD patients suffering from sundowning syndrome[6–8,24] are time-locked to the appearance of pTau in the LPB, and second, that phase delays, and perhaps increased variability, in these rhythms may serve as an early biomarker for the development of pTau in the brainstem and potentially AD more broadly.

**LPB cells expressing dynorphin target well-established circadian structures, the SCN and SPZ, and express pTau in TAPP mice**
Given the strong association between pTau pathology in the LPB and sundowning-relevant circadian dysfunction that we discovered in

TAPP mice, we next wanted to investigate if these effects might be explained by a potential circuit connecting the LPB and the circadian system. We first injected a non-specific retrograde tracer (CTb) into both the SPZ (Fig. 5a) and the SCN (Fig. 5b). After allowing two weeks for the CTb to travel retrogradely, we perfused mice between ZT7 and ZT8 and paired in situ hybridization for *pDyn* with immunohistochemistry for CTb in the LPB. We found that there is a strong projection from LPB$^{dyn}$ neurons to the SPZ with an average of forty CTb-filled cells per LPB section, and forty-two percent of these CTb-labeled cells colocalizing with *pDyn*. We also found that there is a modest projection from LPB$^{dyn}$ cells to the SCN with an average of 12.5 CTb-filled cells per section and 76% of these cells colocalizing with *pDyn*. Overall, we found that about 52% of CTb-filled LPB cells from SCN/SPZ injections were *pDyn*-positive (Fig. 5c, d). There has been some evidence for a similar LPB input to the SCN from CTb retrograde tracing in rats[28], as well as specifically to SCN$^{VIP}$ neurons using conditional monosynaptic retrograde rabies tracing in mice[17], but cells were not quantified nor characterized in these studies and the role of LPB dysfunction in circadian output was not assessed.

Next, we aimed to use a more specific approach to show that LPB$^{dyn}$ cells project to the SCN and the SPZ. We injected a Cre-dependent channelrhodopsin (ChR2) tagged with mCherry into the LPB of *pDyn*-IRES-cre mice and allowed three weeks for the vector to travel anterogradely before perfusing these mice between ZT7 and ZT8. We then conducted immunohistochemistry on hypothalamic sections containing the SCN and SPZ for arginine vasopressin (AVP), a

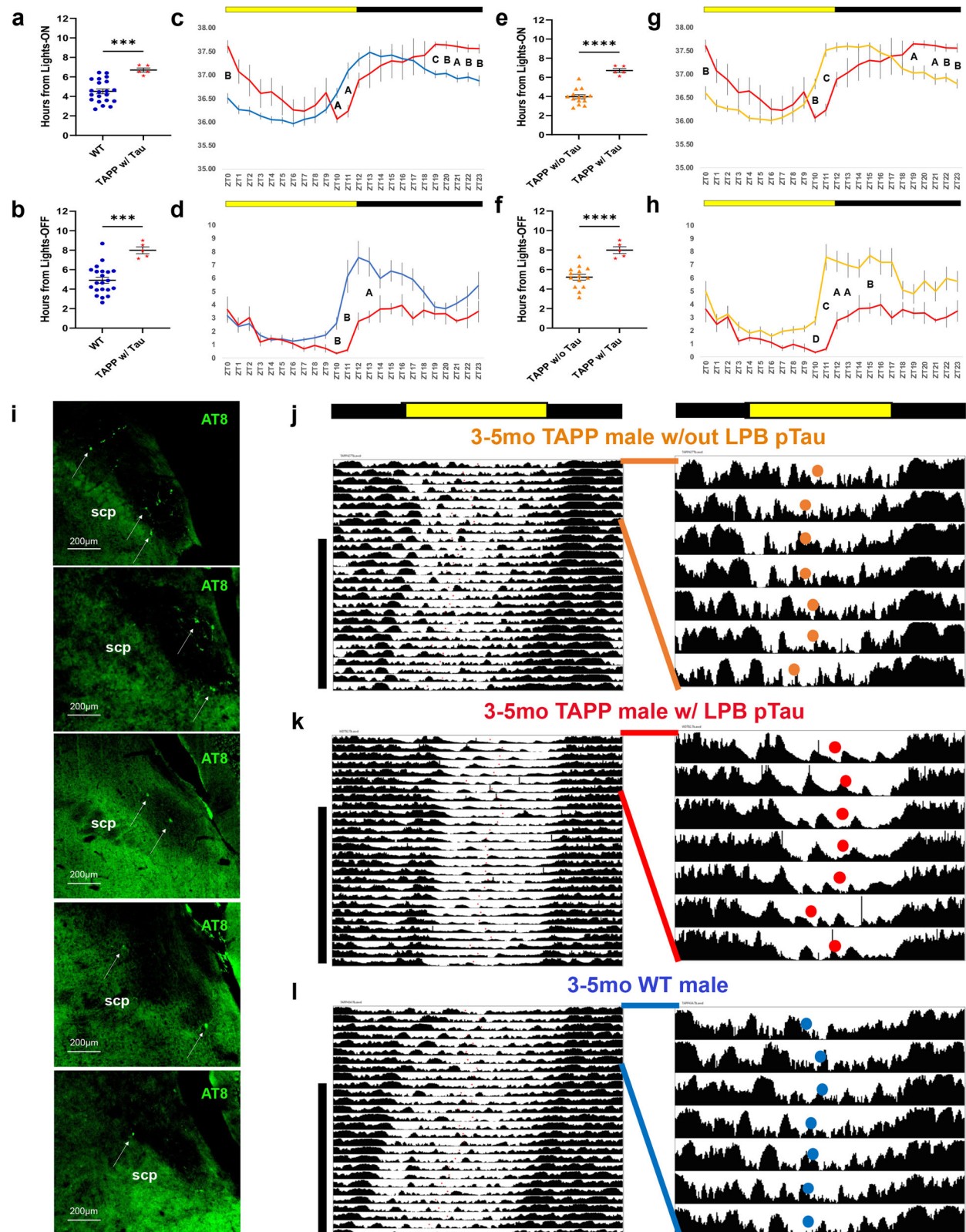

marker that delineates the boundaries for both the dorsal SCN and paraventricular nucleus of the hypothalamus (PVH), thus outlining the ventral and dorsal boundaries of the SPZ, respectively. We found a dense innervation of mCherry-labeled LPB$^{dyn}$ axon terminals throughout the SPZ, and to a lesser extent in the SCN (Fig. 5e), which matches our findings from retrograde tracing with CTb. A previous study by Huang et al. also utilized Cre-dependent anterograde tracing from

LPB$^{dyn}$ neurons and described a "ventral pathway" that supplied numerous hypothalamic areas[29]. Anatomical drawings from that study showed fibers in the SPZ region, but the SPZ was not specifically identified nor were the AVP-expressing neurons that delineate its boundaries examined. Additionally, Huang et al. described little to no labeling in the SCN. Given the relatively few fibers that we found in the SCN compared to the SPZ, it may be that subtle differences in the

**Fig. 4 | 3–5mo TAPP males with LPB pTau exhibit dysfunction of Tb and LMA rhythms. a**, **b** 3–5mo TAPP males with hyperphosphorylated Tau (pTau) in the lateral parabrachial (LPB) (red, $n = 5$) exhibited later body temperature (Tb) bathyphases [Two-tailed unpaired $t$ tests, $t_{(24)} = 4.089$, ***$p = 0.0004$] and locomotor activity (LMA) acrophases [Two-tailed unpaired $t$ tests, $t_{(24)} = 4.501$, ***$p = 0.0001$] than 3–5mo wildtype (WT, blue, $n = 21$ mice) males. **c** 3–5mo TAPP males with LPB pTau (red, $n = 5$ mice) had increased Tb around the active-to-rest phase transition and decreased Tb at ZT10-ZT11 compared to WTs (blue, $n = 21$ mice) [Two-way RM ANOVA, Interaction: $F_{(23,552)} = 4.670$, $p < 0.0001$, Sidak's *post hoc*: ZT0, B$p = 0.0019$, ZT10, A$p = 0.0176$; ZT11 A$p = 0.0230$; ZT19, C$p = 0.0002$; ZT20, B$p = 0.0077$; ZT21, A$p = 0.0225$; ZT22, B$p = 0.0067$; ZT23, B$p = 0.0040$]. **d** 3–5mo TAPP males with LPB pTau (red, $n = 5$ mice) had decreased LMA counts at ZT10-ZT11 and ZT13 compared to WTs (blue, $n = 21$) [Two-way RM ANOVA, Interaction: $F_{(23,552)} = 2.176$, $p = 0.0013$, Sidak's *post hoc*: ZT10, B$p = 0.0031$; ZT11, B$p = 0.0063$; ZT13, A$p = 0.0457$]. **e**, **f** 3–5mo TAPP males with LPB pTau (red stars, $n = 5$ mice) exhibited later Tb bathyphases [Two-tailed unpaired $t$ tests, $t_{(17)} = 7.101$, ****$p < 0.0001$] and LMA acrophases [Two-

tailed unpaired $t$ tests, $t_{(17)} = 5.049$, ****$p < 0.0001$] than 3–5mo TAPP males without LPB pTau (orange triangles, $n = 20$ mice). **g** 3–5mo TAPP males with LPB pTau (red, $n = 5$ mice) had increased active-to-rest phase transition Tb and decreased ZT10-ZT11 Tb compared to those without (orange, $n = 20$) [Two-way RM ANOVA, Interaction: $F_{(23,414)} = 7.443$, $p < 0.0001$, Sidak's *post hoc*: ZT0, B$p = 0.0046$; ZT10, B$p = 0.0022$; ZT11, C$p = 0.0004$; ZT19, A$p = 0.0374$; ZT21, A$p = 0.0172$; ZT22, B$p = 0.0051$; ZT23 B$p = 0.0016$]. **h** 3–5mo TAPP males with LPB pTau showed decreased ZT10-ZT13 and ZT15 LMA compared to those without (orange, $n = 15$ mice) [Two-way RM ANOVA, Interaction: $F_{(23,414)} = 2.571$, $p < 0.0001$, Sidak's *post hoc*: ZT10, D$p < 0.0001$; ZT11, C$p = 0.0001$; ZT12, A$p = 0.0147$; ZT13, A$p = 0.0415$]. **i** Images from all ($n = 5$ mice) 3–5mo TAPP males with LPB pTau (white arrows). **j**, **k**, **l** Left, Tb recordings under 12 h–12 h light-dark (LD) and constant darkness (DD, black vertical bar). Right, magnified LD recordings depicting daily bathyphases as orange (3–5mo TAPP male without LPB pTau), red (3–5mo TAPP male with LPB pTau), or blue (3–5mo WT male). Data are means ± SEM. Source data are provided as a Source Data file. scp: superior cerebellar peduncle.

---

injection sites of that study and our experiments explain this discrepancy. Given the large density of fibers we found in the SPZ, and because the SPZ is the obligate relay of the SCN for numerous physiological rhythms, including Tb and LMA[12], the role of this circuit should be an area of further research moving forward, within and beyond AD.

Finally, we sought to examine whether LPB cells that project to the SCN and SPZ are directly affected by pTau. In male and female TAPP mice we injected CTb into the SPZ and then stained sections containing the LPB from these mice for pTau using the AT8 antibody. We found that a subset of the CTb-filled cells in the LPB indeed express pTau (Fig. 5f). We were surprised that we did not see more of an overlap between CTb and pTau in conjunction with the moderate amount of LPB pTau. One possible explanation for this is the age of the injected mice, which was between 10 and 11mo by the time of injection and subsequent perfusion and staining. Since pTau has been heavily implicated in neurodegeneration, we hypothesized that the LPB^{dyn→SCN/SPZ} pathway may be affected by neurodegeneration.

In order to examine the magnitude to which pTau directly accumulates in LPB^{dyn} cells, we stained a subset of our 7–9mo male TAPP mice for pTau and pDyn. We found that nearly half of pDyn-expressing neurons in the LPB were filled with pTau in 7–9mo TAPP males (Figs. 6a, e, S7). Next, we sought to investigate neurodegeneration in more depth. We aged a subset of male and female TAPP or WT mice to between 12 and 13mo and then again examined for pDyn and pTau expression in the LPB of these aged mice. We noticed that the effect of neurodegeneration in these aged mice was variable both across and within sex. Two of the five males and half of the females showed severe loss of pDyn-expressing cells in the LPB, although the overall level of cell loss seemed to be worse in aged females than in aged males (Figs. 6b–d, S8, S9). These data suggest that the development of pTau in the LPB, and specifically in LPB^{dyn} neurons, leads to neurodegeneration in an age- and sex-dependent manner which disrupts and may eventually destroy the LPB^{dyn→SCN/SPZ} pathway. The effect of neurodegeneration appeared to have progressed faster in females than in males as well, which mirrored the earlier development of pTau in 3–5mo TAPP females compared to 3–5mo TAPP males (Figs. 1 and 2).

We also stained TAPP mice for Aβ pathology in order to investigate how much of these effects could be due to the other hallmark of AD. We found that, similar to pTau, 3–5mo TAPP males do not show consistent Aβ pathology but 7–9mo TAPP males and both 3–5mo and 7–9mo TAPP females do (Fig. S10). We did not consistently find Aβ pathology in lower brain regions such as the brainstem or the hypothalamus at any of these ages, but instead in cortical structures (Fig. S11) and the hippocampus. However, none of these cortical or hippocampal areas were found to project to the SCN or SPZ in our CTb retrograde tracing experiments. Additionally, we closely examined other brain regions known to project to the circadian system, including

the midbrain raphe nuclei and the intergeniculate leaflet, and did not see consistent pTau pathology in these regions that appeared in conjunction with the emergence of circadian dysfunction (Fig. S12). Altogether, the location of Aβ labeling and the lack of pTau development in other circadian input targets at ages when circadian disturbances are detected further supports the role of LPB pTau pathology in the circadian dysfunction seen at these ages.

## TAPP males and females that express pTau in the LPB exhibit increased aggression around the active-to-rest phase transition

In addition to wandering behavior and other disturbances during the late afternoon and early evening (around the active-to-rest phase transition) in AD patients, sundowning is most frequently characterized by agitation and aggression during this time[9,10]. Recently, we established that aggression propensity in mice exhibits a daily rhythm which is dependent on a circadian circuit that includes both the SCN and the SPZ[16]. Notably, disrupting this circuit increased behavioral aggression around the active-to-rest phase transition, suggesting that disturbances to the proper functioning of these structures might underlie the temporal increase in aggression seen during sundowning. We thus sought to explore the possibility that TAPP mice may also exhibit a time-dependent increase in aggression similar to that seen in AD patients with sundowning. We used the resident intruder paradigm to measure behavioral aggression propensity, as in our prior published work[16], at four different time points: one hour before and after the active-to-rest phase transition (ZT23 and ZT1, respectively) and 1 h before and after the rest-to-active phase transition (ZT11 and ZT13, respectively) in male TAPP and WT mice. In 3–5mo WT males, we saw a rhythm of aggression propensity with a trough at ZT1 and a peak at ZT13, matching what we have previously shown in intact male mice[16]. In our 3–5mo TAPP males we saw an arrhythmic phenotype with an apparent increase in aggression at ZT1 compared to WT males (Fig. 7a). We noticed that our data did not follow a normal distribution and when we specifically compared aggression at ZT1 between 3–5mo WT and TAPP males using a non-parametric test we found an increase in aggression in TAPP mice at this time point (Fig. 7b). Additionally, we saw that the same five 3–5mo TAPP males who already exhibited pTau in the LPB displayed increased aggression at ZT1 compared to 3–5mo TAPP males without pTau in the LPB (Fig. 7b–e). In 7–9mo males we saw an increase in aggression at ZT23, immediately before lights-on (Fig. 7f, g). Additionally, the TAPP male with the most aggression was heavily impacted by pTau in LPB^{dyn} cells while the TAPP male with the lowest level of aggression had very little pTau in the LPB (Fig. 7h, i). Given the increased aggression phenotype we observed in TAPP males, we next tested 3–5mo TAPP and WT females at ZT23 using the same resident intruder paradigm, but with juvenile female intruders which has been shown to elicit female aggression[30]. All TAPP females we examined show LPB pTau at this age, and similar to 7–9mo TAPP

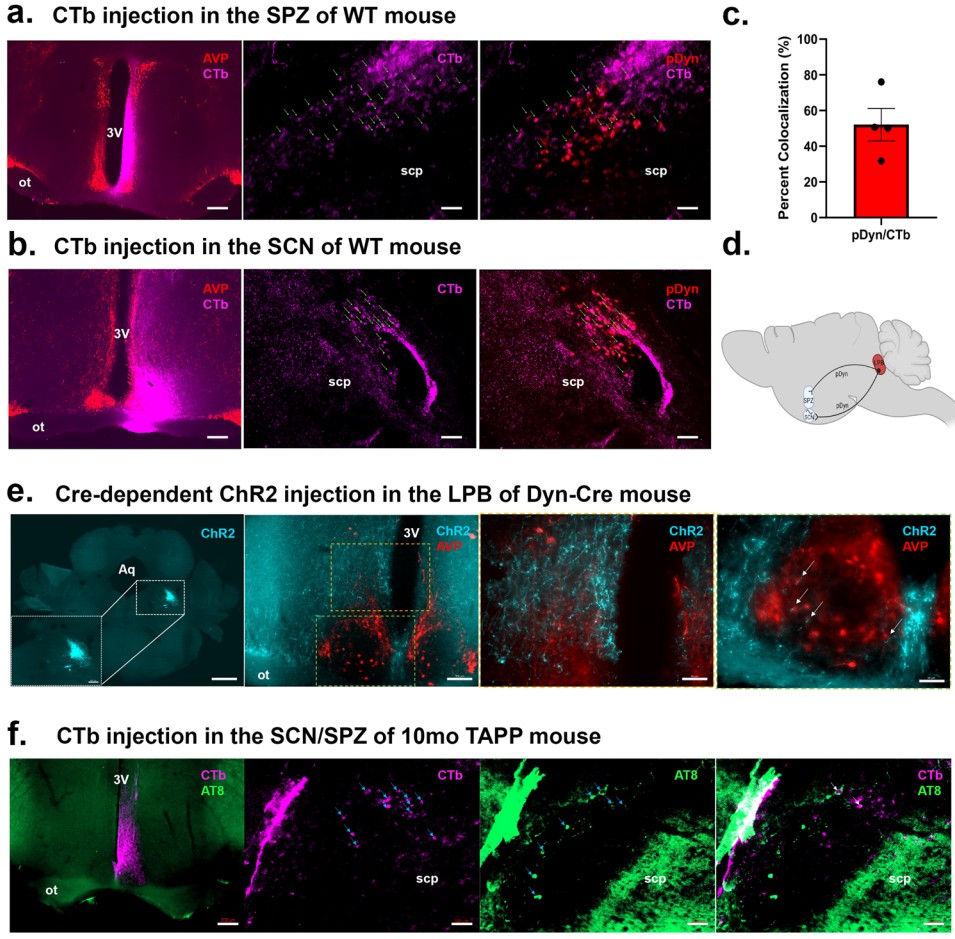

**a.** CTb injection in the SPZ of WT mouse

**b.** CTb injection in the SCN of WT mouse

**c.**

**d.**

**e.** Cre-dependent ChR2 injection in the LPB of Dyn-Cre mouse

**f.** CTb injection in the SCN/SPZ of 10mo TAPP mouse

**Fig. 5 | Dynorphin cells in the LPB project to both the SCN and the SPZ, revealed by retrograde and anterograde tracing. a** Left, injection in WT mouse of fluorescently conjugated retrograde tracer, cholera toxin subunit b (CTb, hot pink), into the subparaventricular zone (SPZ), delineated ventrally by arginine vasopressin (AVP, red)-expressing neurons of the suprachiasmatic nucleus (SCN) and dorsally by AVP-expressing neurons of the paraventricular nucleus. Scale bar = 200 μm. Middle, numerous CTb-labeled cell bodies (small blue arrows) are revealed in the lateral parabrachial nucleus (LPB). Scale bar = 50 μm. Right, several of these CTb-labeled LPB neurons co-localize (small blue arrows) with *pDyn* (red, indicative of dynorphin-expressing cells) using RNAscope in situ hybridization. Scale bar = 50 μm. *n* = 4 male mice. **b** Left (scale bar = 200 μm), in separate WT mice, injection of CTb (hot pink) localized within the AVP-expressing (red) boundary of the SCN similarly revealed several CTb-labeled cell bodies in the LPB (middle, small blue arrows, scale bar = 50 μm), many of which were found to also express *pDyn* (bottom, small blue arrows on red neurons, scale bar = 50 μm.). **c** Quantification of CTB/*pDyn* overlap in the LPB of WT mice with CTb injected into the SCN/SPZ (*n* = 4 males) showing 52% ± 9% *pDyn*/CTb overlap. **d** Schematic of LPB^dyn→SCN/SPZ projection. Created with Biorender.com. **e** Left, genetically-targeted anterograde tracing from LPB^dyn neurons using *pDyn*-IRES-Cre mice and Cre-dependent tracer (AAV8-hSyn-DIO-ChR2-mCherry, turquoise, scale bars: low-magnification = 500 μm, high-magnification = 200 μm) reveals fibers and boutons in the SPZ (top box in second from left, and second from right) and to a lesser extent the SCN [bottom box in second from left (scale bar = 50 μm), and right (scale bar = 50 μm)]. AVP (red) labeling denotes dorsal boundary of the SCN. *n* = 4 male mice. **f** Left, CTb injection (hot pink, scale bar = 200 μm) into the SPZ of 10mo TAPP mouse reveals CTb-labeled LPB neurons (second from left, blue arrows, scale bar = 50 μm) some of which co-localize (white arrows in right, scale bar = 50 μm) with pTau (green = AT8 antibody, second from right and right, scale bar = 50 μm). *n* = 3, 2 females and 1 male mice. Data presented as means ± SEM. Source data are provided as a Source Data file. 3 V, third ventricle. ot, optic tract. scp, superior cerebellar peduncle. Aq, cerebral aqueduct.

males, we found increased aggression propensity in TAPP females compared to WT females at ZT23 (Fig. 7j). We have previously shown that aggression propensity at ZT1 increases when the SPZ is disrupted in males of similar ages to our young TAPP mice used here[16]. Given the projection from the LPB to the SCN and SPZ (Fig. 5), and that increased aggression in TAPP males is only seen when pTau is present in the LPB (Fig. 7e), this LPB^→SCN/SPZ circuit is a promising candidate for the source of circadian-mediated increases in aggression and agitation seen in AD patients struggling with sundowning syndrome.

**Genetically targeted ablation of LPB^dyn cells partially recapitulates the phenotypes seen in TAPP mice in a sex-dependent manner**

Since we saw striking cell death in some TAPP mice at 12–13mo, as well as pTau in nearly half of LPB^dyn neurons prior to such cell death (Fig. 5),

we sought to examine the effects of killing LPB^dyn neurons in otherwise intact mice. We used genetically targeted ablations via a Cre-dependent caspase vector (AAV5-flex-taCasp3-TEVp) to kill LPB^dyn cells in *pDyn*-IRES-cre mice and compared Tb and LMA rhythms before and after these injections. We recorded these mice for five weeks post-injection and compared pre-condition rhythms to those at two weeks post-injection. In male LPB caspase-injected *pDyn*-cre mice, we found a striking hyperthermic response at nearly every time point at two weeks post-injection (Fig. 8a) as well as a stark increase in area under the curve (AUC, Fig. 8b). Interestingly, we did not find any major changes to LMA rhythms upon ablation of LPB^dyn cells nor did we see a change in LMA AUC (Fig. 8c, d). In females we found a very similar general hyperthermic response at two weeks post-injection, although there were relatively fewer time points that reached significance (Fig. 8e). Additionally, there was a concurrent increase in

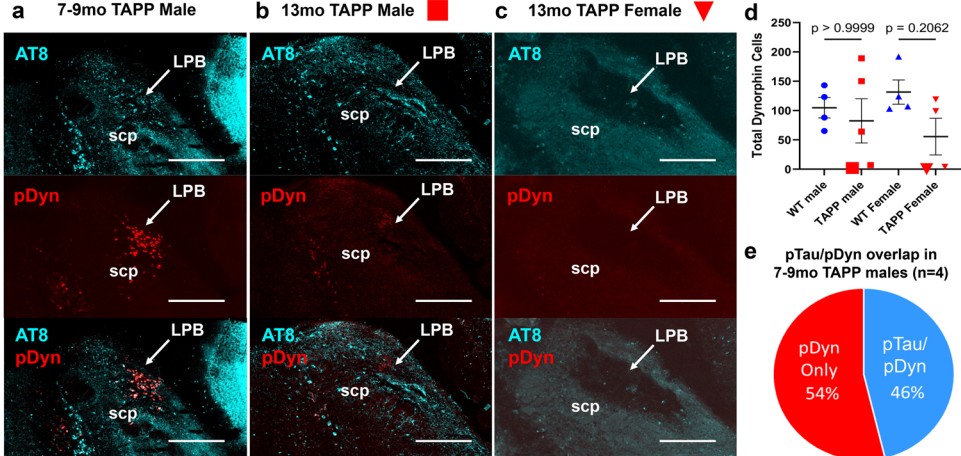

**Fig. 6 | Appearance of Tau tangles in LPB dynorphin neurons precedes cell death in some TAPP mice, and at a higher percentage in TAPP females.**
**a** Representative image depicting the co-localization of pTau (turquoise = AT8 antibody, top and bottom) and *pDyn* (red = RNAscope in situ hybridization for *pDyn*, middle and bottom) in the LPB (*n* = 4 mice). Co-localization is revealed by white cell bodies (bottom). Scale bars = 200 μm. **b, c** At older ages (12–13mo) some male and female TAPP mice exhibit a loss of *pDyn*-expressing (middle and bottom) LPB neurons following such pTau pathology (top and bottom) (*n* = 2 males, *n* = 2 females). Scale bars = 200 μm. **d** However the overall numbers of *pDyn* LPB neurons was not lower across male (red squares, *n* = 5) and female (red triangles, *n* = 4) TAPP mice compared to male (blue circles, *n* = 4) and female (blue triangles, *n* = 4) WT

mice, despite TAPP females representing a higher percentage of mice showing a loss of LPB dynorphin cells compared to TAPP males [One-way ANOVA with planned comparisons between WT and TAPP males and WT and TAPP females with Bonferroni's correction, $F_{(3,13)} = 1.123$ $p = 0.3755$, *post hoc*; WT male vs TAPP male $t_{(13)} = 0.5434$, $^{ns}p > 0.9999$, WT female vs TAPP female $t_{(13)} = 1.753$, $^{ns}p = 0.2062$]. Additionally, it should be noted that multiple 12–13mo TAPP females showed apparent cell loss in the form of a lesion visible in LPB region, as depicted in (**c**). **e** Quantification of co-localization in 7–9mo TAPP mice (*n* = 4) revealed that 46% ± 5% of *pDyn*-expressing LPB neurons also exhibit pTau at 7–9 months of age. Data presented as means ± SEM. Source data are provided as a Source Data file. scp, superior cerebellar peduncle.

AUC for the Tb rhythm (Fig. 8f). In females we also did not see a significant change in the LMA rhythm at two weeks post-injection (Fig. 8g), nor any change in AUC for the LMA rhythm (Fig. 8h). The increase in Tb AUC after caspase-mediated ablation of LPB^dyn neurons mimics that of 7–9mo male and female TAPP mice (Fig. S13) perhaps indicating that LPB^dyn cells in these TAPP mice are beginning to degenerate. Finally, following perfusion of LPB caspase-ablated *pDyn*-cre mice, we conducted in situ hybridization for *pDyn* in the LPB and saw that our ablations successfully killed these cells (Fig. 8i, j). We quantified a 91% reduction in LPB^dyn cells when comparing caspase ablated mice to WT controls (Fig. 8k), and the mice who had fewer remaining LPB^dyn cells experienced more severe hyperthermia as measured by increases in AUC of the Tb rhythm (Fig. 8l). It should also be noted that the three *pDyn*-cre mice with the most remaining LPB^dyn cells were all female (pink dots in Fig. 8m), and this may at least partially explain why the general hyperthermia was less severe in ablated females.

We also compared Tb and LMA rhythms at four weeks post-injection to examine whether these mice could use some compensatory mechanism to limit the ablation-induced hyperthermia (Fig. S14). We found that males were still drastically hyperthermic at numerous time points with increased AUC (Fig. S14a, b). We also analyzed Tb bathyphase during the pre-injection period and at both two- and four-weeks post-injection (Fig. S14c). We found a slight phase advance at two weeks and no change at four weeks. For LMA we again found that the rhythm was largely unaffected by LPB^dyn ablation, however, there was a decrease at ZT12 with no change in AUC or LMA acrophase (Fig. S14d–f). For females there was again an increase in Tb at similar time points to those seen at two weeks post-injection and an increase in AUC (Fig. S14g, h). There was also a slight phase advance in Tb bathyphase for LPB^dyn ablated mice at both two-and four-weeks post-injection (Fig. S14i). For LMA in females we saw a decrease at ZT13 and a slight dip in AUC with no change in LMA acrophase (Fig. S14j–l). In order to demonstrate the effectiveness of our in-situ protocol we also included an image of a *pDyn*-cre mouse that received bilateral caspase injections but only experienced

complete LPB^dyn cell loss unilaterally, with partial loss on the other side (Fig. S14m, n).

It was interesting that we did not see a hyperthermic response at the time points surrounding the active-to-rest phase transition in LPB^dyn ablated mice. There are a few potential explanations for this. First, the observed hyperthermia may be so stark throughout the light and dark phase that as mice transition into sleep or quiet rest a compensatory mechanism, perhaps the circadian system itself, does not allow Tb to remain as elevated. Secondly, our ablations were very effective with near total LPB^dyn cell loss, and as such, this could be more analogous to what would be seen in TAPP mice aged well beyond 7–9mo. When pairing this with what we showed in our hr-by-hr graphs in Fig. 2, it becomes clear that the 7–9mo TAPP males and females both show hyperthermia much more similar to what is shown by our caspase-mediated ablations. This would indicate that the death of LPB^dyn neurons leads to general hyperthermia throughout the day once enough of those cells are lost, analogous to what we show in Figs. 8, S14. In contrast, when the LPB^dyn→SCN/SPZ pathway is affected by pTau (prior to neurodegeneration) the hyperthermia is mostly confined to the active-to-rest phase transition and the phase delay predominates. General hyperthermia of this sort has indeed been shown in AD patients[6]. This also would offer an explanation as to why LPB^dyn caspase-mediated ablations caused an apparent phase advance in Tb rhythms. After ablation of LPB^dyn cells the Tb values later in the rest phase are elevated to a greater extent compared to those in the very early rest phase. This pushes the Tb bathyphase value earlier which is read as an advance. This hyperthermia may be so strong that it, in effect, masks any underlying circadian changes that ablating LPB^dyn cells may cause.

Lastly, it is well-established that circadian output weakens with age[7,31–33], and so the effects of pTau on the circadian system through the LPB^dyn→SCN/SPZ pathway may be worsened by the process of natural aging in addition to the accumulation of pTau pathology itself. The mice that received caspase-mediated ablation of LPB^dyn neurons were at 3–5mo, and so they would presumably have a stronger circadian influence on Tb than mice at 7–9mo or older.

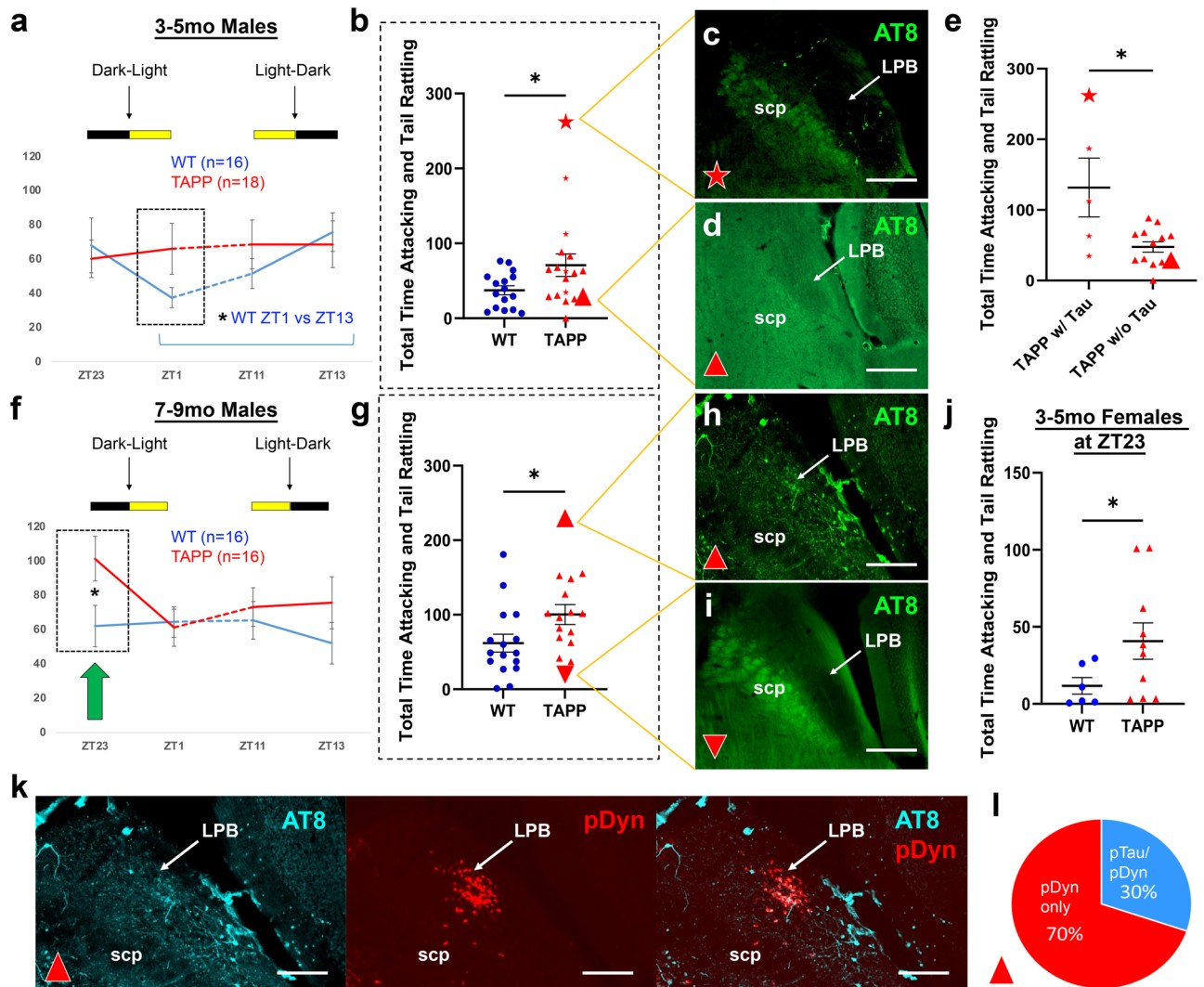

**Fig. 7 | TAPP mice with LPB pTau exhibit increased aggression around the active-to-rest transition. a** 3–5mo wildtype (WT) males (blue, $n = 16$) showed aggression rhythms (total time attacking and tail rattling, in s) with troughs at zeitgeber time (ZT)1 (lights-on=ZT0, lights-off=ZT12) and peaks at ZT13, as seen previously[16] [Planned comparison, One-way RM ANOVA, $F_{(3,45)} = 3.927$, $p = 0.0143$, Tukey's multiple comparisons *post hoc*: $p = 0.0155$]. This rhythm was blunted in 3–5mo TAPP males (red, $n = 18$) [Planned comparison, One-way RM ANOVA, $F_{(2.629,44.69)} = 0.2921$, $p = 0.8051$]. **b** Planned comparison at ZT1 revealed significantly increased aggression for all 3–5mo TAPP males (red, $n = 18$) compared to WTs (blue, $n = 16$) [One-tailed Mann–Whitney $U$ test, $U = 92$, *$p = 0.0377$]. Five 3–5mo TAPP males exhibiting lateral parabrachial (LPB) hyperphosphorylated Tau (pTau, red stars, see **c**, green=AT8 antibody) were among the most aggressive at ZT1 compared to those without (red triangles, $n = 13$, see **d**). Scale bars=200 μm. **e** ZT1 Planned comparison between 3–5mo TAPP males with LPB pTau (red stars, $n = 5$) revealed significantly increased aggression compared to those without (red triangles, $n = 13$) [One-tailed Mann–Whitney $U$, $U = 12$, *$p = 0.0230$]. **f** 7–9mo WT

males (blue, $n = 16$) showed age-related blunting of this rhythm [Planned comparison, One-way RM ANOVA, $F_{(2.287,34.31)} = 0.5174$, $p = 0.6247$], whereas TAPP males exhibited increased aggression at ZT23 (green arrow) [Planned comparison, see (**g**)]. **g** ZT23 Planned comparison between 7 and 9mo TAPP (red, $n = 16$) and WT (blue, $n = 16$) males [Two-tailed Mann–Whitney $U$, $U = 69$, *$p = 0.0260$] in (**f**). The most aggressive 7–9mo TAPP male at ZT23 exhibited heavy LPB pTau (large red triangle, see **h**) compared to the least aggressive (large upside-down red triangle, see **i**). Scale bars = 200 μm. **j** 3–5mo TAPP females (red, $n = 10$) also exhibited increased ZT23 aggression compared to WT females (blue, $n = 6$) [Two-tailed Mann–Whitney $U$, $U = 11$, *$p = 0.042$]. **k** The most aggressive 7–9mo TAPP male in (**g**) (large red triangle) showed much overlap (white, right) between pTau (turquoise = AT8 antibody, left and right) and *pDyn* (red, middle and right) in LPB neurons. Scale bars = 200 μm. $n = 4$ 7–9mo TAPP male mice. **l** 30% of *pDyn*-expressing LPB neurons exhibited pTau in the mouse depicted in (**k**). Data are means ± SEM. Source data are provided as a Source Data file. scp, superior cerebellar peduncle.

## Discussion

Here we have shown that LPB$^{dyn}$ cells send projections to the SCN and SPZ (Fig. 5), structures which are known to mediate circadian rhythms of Tb, LMA, and aggression[12,16,17]. AD patients frequently suffer from sundowning syndrome which is characterized by agitation, aggression, and wandering during the late afternoon and early evening, the transition between their active and rest phases[9,10], as well as a phase delay in Tb and LMA rhythms[6–8]. This implies that the SCN and SPZ are likely adversely impacted by AD pathogenesis, however recent work examining actigraphy combined with hypothalamic tissue showed no

difference in SCN$^{VIP}$ cell loss between phase-delayed AD patients and healthy age-matched controls who were not phase-delayed[7,18]. This led to the idea that a dysfunctional input to the circadian system might underlie circadian alterations in AD patients. We demonstrate that TAPP mice develop a phase delay similar to that reported in AD patients, and that such circadian dysfunction arises in conjunction with pTau in the LPB in a sex-dependent manner. Our current work also heavily implicates LPB$^{dyn}$ cells as an important input to the SCN/SPZ that is extensively impacted by pTau accumulation and neurodegeneration (Figs. 5, 6). We show that increases in Tb, LMA, and aggression

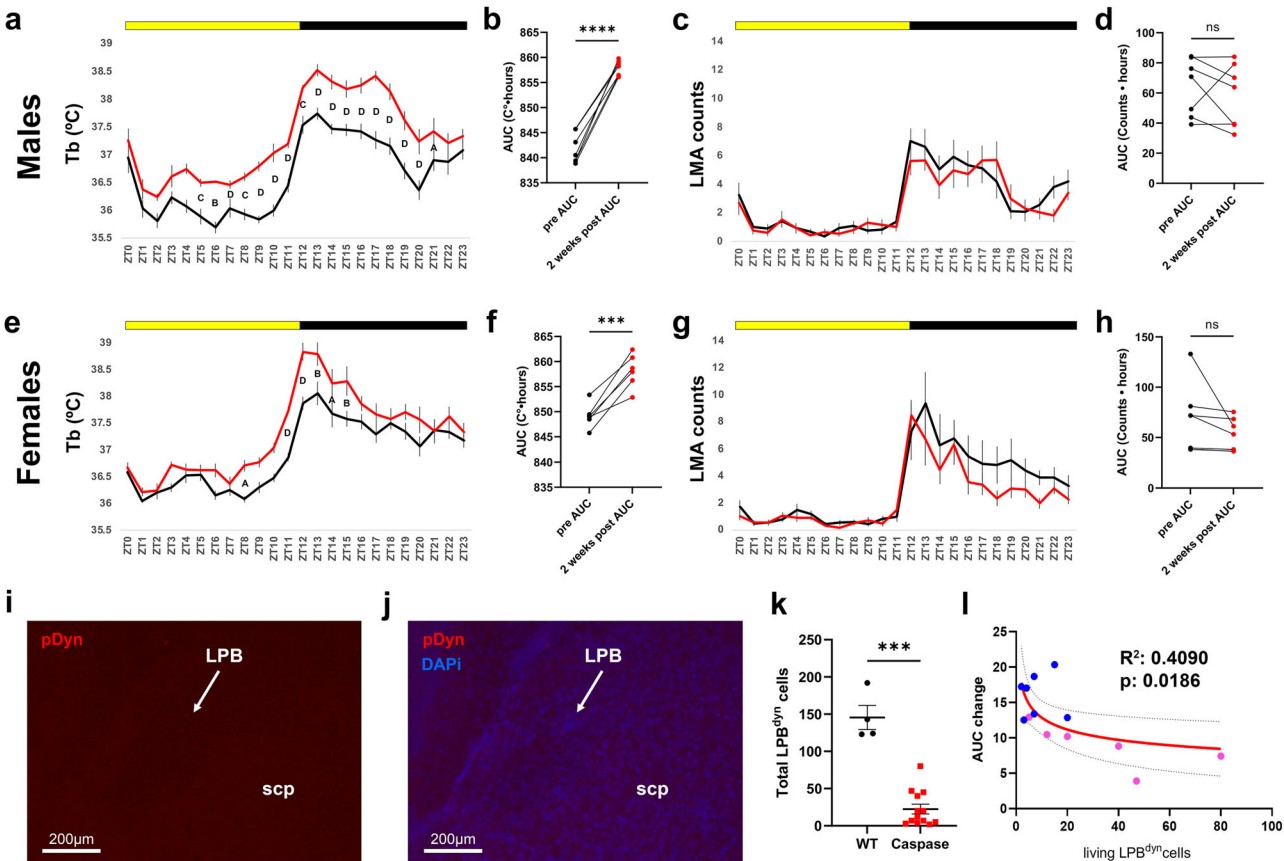

**Fig. 8 | Genetically targeted ablation of LPB$^{dyn}$ cells via caspase in *pDyn*-IRES-Cre mice increases Tb. a** Lateral parabrachial dynorphin (LPB$^{dyn}$)-ablated males ($n = 7$; red, post-injection; black, pre-injection) exhibited increased body temperature (Tb) throughout the light and dark phases two weeks post-injection of caspase [two-way RM ANOVA, Interaction: $F_{(23,138)} = 3$, $p < 0.0001$, Sidak's *post hoc*: ZT4, $^{C}p = 0.0002$; ZT5, $^{B}p = 0.0015$; ZT6, $^{D}p < 0.0001$; ZT8, $^{C}p = 0.0002$; ZT9, $^{D}p < 0.0001$; ZT10, $^{D}p < 0.0001$; ZT11, $^{D}p < 0.0001$; ZT12, $^{C}p = 0.0003$; ZT13, $^{D}p < 0.0001$; ZT14, $^{D}p < 0.0001$; ZT15, $^{D}p < 0.0001$; ZT16, $^{D}p < 0.0001$; ZT17, $^{D}p < 0.0001$; ZT18-ZT20, $^{D}p < 0.0001$; ZT21, $^{A}p = 0.0155$]. **b** Area under the curve (AUC) increased from pre- (black) to post-injection (red) recordings ($n = 7$) [Two-tailed paired *t* test, $t_{(6)} = 13.7$, ****$p < 0.0001$] (**c**). LPB$^{dyn}$-ablated males ($n = 7$; red, post-injection; black, pre-injection) showed no locomotor activity (LMA) differences two weeks post-injection [Two-way RM ANOVA, Interaction: $F_{(23,138)} = 1.101$, $p = 0.3515$, Sidak's *post hoc*]. **d** LPB$^{dyn}$-ablated males ($n = 7$; red, post-injection; black, pre-injection) showed no LMA AUC differences [two-tailed paired *t* test, $t_{(6)} = 0.7746$, $^{ns}p = 0.468$]. **e** LPB$^{dyn}$-ablated females ($n = 6$; red, post-injection; black, pre-injection) showed increased Tb at multiple timepoints throughout the light and

dark phases [two-way RM ANOVA, Interaction: $F_{(23,115)} = 2.387$, $p = 0.0013$, Sidak's *post hoc*: ZT8, $^{A}p = 0.0158$; ZT11, $^{D}p < 0.0001$; ZT12, $^{D}p < 0.0001$; ZT13, $^{B}p = 0.19$; ZT14, $^{A}p = 0.0439$; ZT15, $^{B}p = 0.0031$]. **f** LPB$^{dyn}$-ablated females had increased Tb AUC two weeks post-injection [Two-tailed paired *t* test, $t_{(5)} = 7.117$, ***$p = 0.0008$]. **g** LPB$^{dyn}$-ablated females ($n = 6$; red, post-injection; black, pre-injection) did not exhibit LMA changes [Two-way RM ANOVA, Interaction: $F_{(23,115)} = 1.176$, $p = 0.2807$, Sidak's *post hoc*: all ns]. **h** LPB$^{dyn}$-ablated females ($n = 6$; red, post-injection; black, pre-injection) showed no LMA AUC difference in AUC [Two-tailed paired *t* test, $t(5) = 1.531$, $^{ns}p = 0.1863$]. **i**, **j** Example LPB section with dynorphin cells (red, *pDyn* using RNAscope) ablated and counterstained with DAPi (blue). $n = 13$. White arrows denote the LPB. **k** Caspase injections (red, $n = 13$) led to 91% reduction in LPB$^{dyn}$ cells compared to WTs ($n = 4$, 2 males and 2 females at 3–5mo) [Mann–Whitney *U* test, $U = 0$ ***$p = 0.0008$]. **l** Mice with more LPB$^{dyn}$ cells after injection experienced less of an Tb AUC ($n = 6$ females, pink; $n = 7$ males, blue) [log–log regression with *F* test: $F_{(1,11)} = 7.613$, $\beta_1 = -0.2021$, $p = 0.0186$, R$^2 = 0.4090$]. Data are means ± SEM, except (**l**) which is mean ± 95% CI. Source data provided as a Source Data file. scp, superior cerebellar peduncle.

---

around the active-to-rest phase transition (dark-to-light, the late afternoon for nocturnal mice), as well as the phase delays in Tb and LMA rhythms, are present in TAPP mice only when pTau has accumulated in the LPB, but not before (Figs. 1,2 and 6). Additionally, when we ablated LPB$^{dyn}$ cells we partially recapitulated the hyperthermic Tb phenotype seen in TAPP mice with LPB pTau, particularly at later ages (Fig. 8).

The immediate outputs of the SCN and SPZ that mediate rhythms of LMA and aggression are the DMH and VMH, respectively[12,16,17,34], and Tb rhythms have been shown to be mediated by the SCN and SPZ through an unknown downstream pathway[12,15]. This, paired with the anatomical and behavioral data presented in this paper, has led us to a model (Fig. 9) wherein disruption of the LPB$^{dyn \rightarrow SCN/SPZ}$ pathway by pTau pathology disturbs the normal rhythmicity produced by these circadian structures and leads to sundowning-like symptoms. Thus, altered output to the DMH could lead to phase delayed LMA rhythms and the

temporal increase in LMA that underlies wandering, while altered output to the VMH could lead to the temporal increase in aggression, both of which characterize sundowning syndrome. Since the SCN and SPZ also regulate rhythms of Tb by a currently unknown downstream pathway[12,15], such LPB$^{dyn \rightarrow SCN/SPZ}$ dysfunction could also underlie the phase delay in Tb that has been shown to be highly associated with sundowning[24].

Circadian rhythm changes have been shown in patients with preclinical AD[35], and are hypothesized to be causative in the worsening of AD pathology[36]. Circadian alterations also change across the course of the disease, with patients in preclinical stages exhibiting disrupted sleep and melatonin rhythms[37,38], as well as fragmented activity rhythms, which worsen with increased pathology[35]. Symptomatic AD patients commonly exhibit a phase delay[8], which can be seen in both Tb and LMA[6,7]. Despite the existing data on circadian deficits over the course of AD, most existing AD mouse models fail to recapitulate such

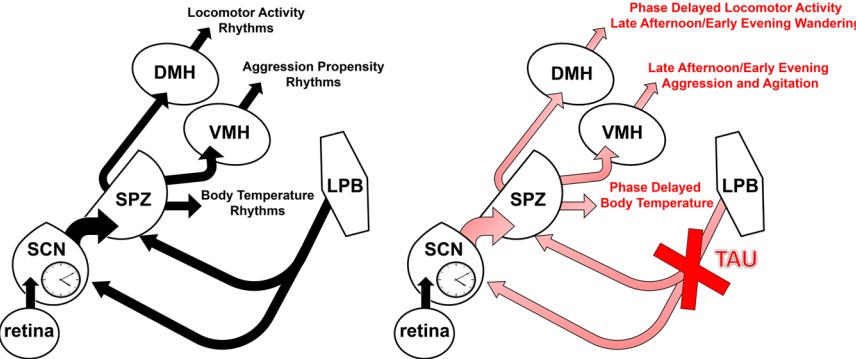

**Fig. 9 | Potential neural circuit underlying sundowning-related symptoms in Alzheimer's disease based on our current findings.** Tau pathology in lateral parabrachial (LPB) dynorphin neurons that project to the suprachiasmatic nucleus (SCN) and subparaventricular zone (SPZ) could cause dysfunction in the SCN→SPZ output pathways that are known to regulate body temperature (Tb) (through a yet to be determined pathway), locomotor activity (LMA) (through the dorsomedial hypothalamus, DMH) and aggression (through the ventromedial hypothalamus, VMH). Such pTau-related LPB dysfunction could thus result in the phase delay in Tb, the phase delay in LMA and temporal increases in wandering through a dysfunctional SCN→SPZ→DMH circuit, as well as the time-dependent increase in aggression through a dysfunctional SCN→SPZ→VMH circuit.

disturbances, particularly those related to sundowning. As recently reviewed by Sheehan and Musiek[8], there exists a wide array of phenotypes expressed across the AD mouse models in which circadian function has been assessed. There have been comparably far more circadian studies on APP/ human presenilin 1 (PS1) mouse models than on Tau transgenic lines and fewer still on lines that contain both mutated Tau and Aβ. Broadly, mice that exhibit Aβ plaque formation via human APP and human PS1, but not pTau neurofibrillary tangles, exhibit varying circadian alterations but do not reliably recapitulate the neurodegeneration nor the specific circadian dysfunction seen in AD patients. Instead, transgenic models expressing pTau pathology due to overexpression of the human MAPT gene, specifically harboring either a P301S or P301L mutation, do show striking neurodegeneration and often more similarly recapitulate the circadian phenotypes that are seen in AD patients[8]. For example, the triple transgenic model (3XTgAD), expressing both Aβ and pTau pathology, has shown circadian phenotypes closer to those observed in AD patients; however, these have been inconsistent. A double knock-out model of the ApoE4 gene has also shown some circadian alterations[39], however the relevance to human AD of these models is unclear because the risk factor associated with this gene in humans is not a knockdown of ApoE4, but rather presence of the ε4 allele of the gene. When pairing APP with the human ε4 allele of ApoE circadian alterations in wheel running are observed[40,41]. Crucially, these alterations were not worse in mice containing both APP and ε4 than in those containing only APP, despite the fact that combining these genotypes does produce increased Aβ deposition[42]. Finally, the PLB1 model, which expresses APP, Tau, and PS1 only in the forebrain did not exhibit any circadian alterations[43]. This led to the idea that such circadian disruptions arise from pathology in the brainstem, and further supports the relevance of pTau in the brainstem LPB→SCN/SPZ neurons in pathogenesis of circadian dysfunction.

Together, the preclinical data from various AD mouse models seems to indicate that models exhibiting pTau aggregation recapitulate circadian alterations that are seen in AD patients better than other models, especially when those models also express mutant Aβ. The contribution of Aβ pathology directly to circadian disruption is unclear and the impact of AD pathology on circadian disruption seems to be coming from an input to the circadian system instead of directly from the SCN/SPZ. Here, we show that there is a circuit connecting the LPB, an important relay station for temperature, sleep/wake, and locomotion, to both the SCN and SPZ. Furthermore, this pathway is directly affected by pTau and the appearance of this pathology in the LPB coincides with the appearance of both phase delays and increased Tb

and LMA at the active-to-rest phase transition. Finally, the number of LPB[dyn] neurons is reduced in some TAPP mice as compared to age-matched controls suggesting that this pathway eventually degenerates with age and the prolonged presence of pTau pathology. Altogether we show a clear relationship between AD pathology in a specific brainstem nucleus that projects to the circadian system and the appearance of circadian deficits.

We also show that the development of such AD pathology coincides with increased behavioral aggression around the active-to-rest phase transition, which is temporally consistent with when AD patients display aggression and agitation during sundowning. We previously showed that, in mice who no longer have the ability to release GABA from the SPZ, the total time attacking increases at ZT1[16]. This increase in aggression appears to result from disruption of a functional connection between the SPZ and VMH neurons that promote aggression. In our current study, we showed that mice exhibiting pTau in the LPB show a similar increase in total time attacking around the active-to-rest phase transition – first during the very early stages of such pathology at ZT1, and then more significantly at ZT23 when this pathology is pronounced in nearly all TAPP mice. Additionally, we show in these mice that LPB neurons, and in particular LPB[dyn] neurons, exhibit pTau and strongly send projections to the SPZ (Figs. 5, 7k, l). Interestingly, neurons in the LPB located in the region that expresses dynorphin have been shown to be active when rodents are perfused shortly after lights on, strongly suggesting that LPB[dyn] are active around the active-to-rest phase transition when symptoms of sundowning are commonly seen[44,45]. Because these cells not only project to the SCN/SPZ, but also appear to be normally active at this time of day, disruption of this pathway provides a temporal and circuit basis for the increased aggression seen in our mice at ZT23 as well as in AD patients during their transition from the active phase to their rest phase.

Sex differences in AD have been well-characterized in human patients, but relatively less attention has been paid to sexual dimorphism in AD in preclinical research. Given that two-thirds of patients diagnosed with AD are women and that both cognitive decline and sundowning syndrome are more prevalent in women[9,21], investigating how circadian rhythms and AD interact with one another in both males and females is an important question. Women also seem to be more at risk of developing sporadic AD than men when they are either heterozygous or homozygous for the APOE4 allele[46]. One recent study found that the neuroimmune expression profile in the hippocampus of female mice and humans changes over the course of aging[47], and to a lesser extent in males. In 9mo female mice there was a significant upregulation in genes correlated with disease-associated microglia,

including ApoE, providing a potential mechanism for how age, sex, and AD could interact. Another recent study found that rising FSH levels in ovarectomized mice, a model relevant to menopause, led to significantly worse Aβ and pTau accumulation in the hippocampus and cortex of 3XTgAD mice. Notably, this effect was blocked by administration of an FSHβ antibody intraperitoneally every two days for 8 weeks[48]. Prior evidence also showed that the LPB region exhibits pTau pathology and neurodegeneration in AD patients[25–27], with such pathology emerging in the preclinical stages of the disease[25]. Interestingly, the patients in this study affected by LPB pTau during pre-clinical stages were all women, lending further credence to the sexually dimorphic nature of LPB pTau development. In the present study we have shown that female mice develop pTau in the LPB earlier than males, and the concurrent circadian alterations temporally coincide with the development of this pathology. Altogether these data suggest that both aging and sex play a role in pre-symptomatic and symptomatic AD pathogenesis as well as plaque and tangle formation, and that the development of pTau leads to circadian alterations analogous to those seen in AD patients with sundowning syndrome, also in an age- and sex-dependent manner.

Nocturnal issues are among the most common reason for institutionalization of elderly people[49], the most common of these being sleeplessness and pain. In addition to rhythms of LMA and Tb, sleep-wake is under strong circadian control, and sleep-wake is also regulated by the SCN→SPZ→DMH pathway similar to LMA. Thus, the results presented here may also lead to a better understanding of the sleep disturbances that are prevalent in AD patients[3,36,50]. The LPB is known to serve as a relay for numerous physiological signals, including temperature sensation[44], LMA[51], and sleep/wake[52]. Additionally, LPBdyn neurons have been shown to decrease Tb[44,51,53] and LMA[51], as well as drive sleep behavior[54] through separate neural circuitry. For example, LPBdyn cells send projections to the preoptic area of the hypothalamus and stimulation of these terminals has been shown to decrease Tb[51]. This circuit could potentially play a role in the generalized hyperthermia we see at later ages of TAPP mice and in our caspase ablations. Notably, we also implicate the LPB, and LPBdyn neurons, in circadian control. Interestingly, these LPBdyn cells have also been heavily implicated in relaying pain information to the forebrain[55–58], and pain is another common comorbidity of multiple neurodegenerative disorders including AD[59] and Parkinson's disease (PD)[60]. Finally, it is important to stress that the LPB is affected by pathology in AD patients as well as in PD patients[25,61,62] and both of these patient populations suffer from circadian disruptions[63]. Taken together, these data indicate that the LPBdyn→SCN/SPZ pathway may regulate circadian disruptions in other neurodegenerative disorders like PD as well as in pain disorders of patients both with and without neurodegeneration, in addition to its potential role in sundowning syndrome.

Both pTau and Aβ have important roles in AD and specifically in the sleep disturbances observed in AD[50,63]. However, drugs that successfully remove Aβ plaques have not yet proved effective in the treatment of AD, and this has led to the belief that Aβ accumulation could be compensatory instead of causative[64]. Tau removal has received comparably less focus in the clinical treatment of AD, and there are multiple Aβ-independent mechanisms for pTau accumulation[65]. Notably, pTau accumulation has also been much more heavily implicated in neurodegeneration than Aβ in AD patients and AD mouse models[8,66,67]. Given this, it will be important to dedicate more future research to the role of pTau in AD pathogenesis, specifically in the alterations of circadian rhythms. Indeed, as we show here, accumulation of pTau in the LPB→SCN/SPZ circuit is temporally correlated with development of clinically analogous AD circadian symptoms. Tau has been shown to be able to silence neurons in AD-model mice even before accumulating in tangles[68], so circadian dysfunction could be an early warning sign of pTau tangles in the LPB→SCN/SPZ circuit. With this knowledge, the clinical observation of phase delays and increases in Tb

and LMA at the active-to-rest phase transition may indicate pTau accumulation in the LPB and monitoring these rhythms could provide clinicians with a tool for early detection of pTau pathology. Beyond this, alterations in circadian rhythms driven by pTau in the LPB→SCN/SPZ circuit could lead to sleep disturbances and result in decreased glymphatic clearance of Aβ, as has been shown previously[69]. Further research into the circuit mechanisms underlying circadian dysfunction in AD is warranted and could lead to earlier indicators as well as potential treatments for current and future AD patients.

## Methods

All experiments and methods were conducted in accordance with institutional ethical guidelines and were approved by the Institutional Animal Care and Use Committee at the University of Wyoming (Animal Welfare Assurance Number D16-00135).

### Animals

TAPP mice (Taconic, model 2469) on a mixed background and double wild-type (WT) controls on the same genetic background (Taconic, model 3723) were obtained at 4 weeks old. Homozygous *pDyn-IRES*-Cre mice were obtained from Jackson laboratories (strain# 027958) and bred in-house. C57Bl6/J mice were obtained from Jackson laboratories (strain# 000664). Swiss Webster mice were obtained from Charles River and bred in-house. Mice were housed in the vivarium of the Biological Sciences building at the University of Wyoming in standard mouse cages with standard rodent chow (Lab Diet 5001) and water available ad libitum. Mice were housed under a 12 h–12 h light-dark (LD) cycle with lights-on at zeitgeber time 0 (ZT0, 6:00 am/6:00) and lights-off at ZT12 (6:00 pm/18:00), unless housed in constant darkness (DD) as noted.

### Surgery

For biotelemetry recordings of Tb and LMA, as in our prior published work[16,17], mice were first anesthetized with a ketamine (100 mg/kg) and xylazine (10 mg/kg) mixture diluted 1:10 in saline, injected into the intraperitoneal cavity. The belly was shaved and sterilized with alcohol and iodine prep pads and a midline incision was made (1 cm). A biotelemetry transmitter (TAF10, Data Sciences International) was then inserted into the intraperitoneal cavity and the internal membrane was closed with absorbable sutures, followed by the external skin being closed with non-absorbable sutures. Mice were given buprenorphine (0.1 mg/kg) in saline subcutaneously for postoperative analgesia.

For neural injections, as in our prior published work[16,17], mice were placed in a stereotaxic apparatus under anesthesia as described above, in either the same surgery or without transmitter implantation. The scalp was shaved, sterilized with alcohol and iodine prep pads and a midline incision was made (1 cm) to reveal the skull. The skull was cleaned with hydrogen peroxide and burr holes were drilled at the appropriate coordinates over the SCN and SPZ (AP: −0.05 cm, ML: +0.015 cm, DV: −0.51 cm), or over the LPB (AP: −0.54 cm, ML: +0.14 cm, DV: −0.30 cm). A glass micropette filled with the appropriate substance was slowly lowered into the desired area and allowed to settle for 2 min. An air puffer microinjection device (Picospritzer) connected to the micropipette and attached to an air compressor was used to slowly dispense the desired volume. For retrograde tracing from the circadian system, approximately 10–30 nl of cholera toxin subunit B conjugated to a red fluorescent protein (CTb-555, Thermofisher #C34776) was dispensed unilaterally into the SCN/SPZ area in TAPP mice or C57Bl6/J mice. For genetically targeted Cre-dependent anterograde tracing of LPBdyn neurons, approximately 40–100 nl of AAV8-hSyn-DIO-ChR2-mCherry (Addgene, #20297-AAV8) was dispensed unilaterally into the LPB of *pDyn*-IRES-Cre mice. For genetically targeted Cre-dependent ablations of LPBdyn neurons, approximately 40–100 nl of AAV5-flex-taCasp3-TEVp (Addgene #45580-AAV5) was dispensed bilaterally into the LPB of *pDyn*-IRES-Cre mice.

## Biotelemetry recordings

Following surgery, all mice were given at least two weeks to recover and were then moved in their home cages into light-tight and noise-attenuated circadian chambers (Phenome) and given at least 5 days to acclimate before recordings began. Within these chambers, home cages sat atop telemetry receivers connected to a computer and data acquisition system (Data Sciences International, Ponemah software 6.x). Clocklab Chamber Control software (Actimetrics) was used to program the light-dark cycle and also to measure light levels (300 lux during the light period) as well as ambient temperature ($23.5 \pm 1\,°C$) and humidity levels (25–30%). Average Tb and total LMA counts were sampled for 10 s every 5 min for at least 7days under LD conditions. Mice were then transferred into constant darkness (DD) beginning at the end of their normal dark period and maintained in DD for 3 weeks while recordings continued under the same parameters. Over the course of the entire recording period, daily wellness checks were performed using real-time analysis of the collected data from the computer. Food and water levels were checked by visual inspection of the home cage at least twice per week using red light illumination when occurring in the dark phase or under DD conditions. For LD data, daily bathyphases and acrophases were determined using ClockLab Analysis 6 (Actimetrics) and averaged over 7 consecutive days for both Tb and LMA. ClockLab fits the data for each day to a sine wave function of 24 h and determines the trough (bathyphase) and peak (acrophase) of this rhythm according to local time, which we then calculated as hours from lights-on and lights-off, respectively. For each rhythm, hr-by-hr averages during these same 7d were calculated using Excel for Mac (version 16.73). For DD data, the period length for 10 consecutive days was determined using a Chi-squared periodogram in ClockLab Analysis 6 (Actimetrics). These 10d started after 2 weeks in free-running conditions.

## Resident intruder tests

All resident male TAPP mice and WT controls being tested for aggression were sexually experienced and singly housed for several weeks (which reliably produces a territorially aggressive phenotype)[16]. All residents were re-entrained to an LD cycle following DD recordings for at least 2 weeks. Then, a resident intruder test was performed, as in our prior published work[16], at one of 4 timepoints: zeitgeber time 23 (ZT23, ZT0 = lights-on), ZT1, ZT11, or ZT13 – that is, one hour before and after the dark-light and light-dark transitions, respectively. A sexually naïve and group-housed male intruder mouse (C57Bl6/J) was introduced into the resident's home cage and behavioral interactions were video recorded under dim red light (which does not alter the phase of the circadian system) for 10 min. Following 3–5 days of rest, the resident was then tested at the next time point in similar fashion, and this continued until all four timepoints were tested for each resident. The order of tests for each mouse was counterbalanced using a Latin square design, with mice randomly assigned to each order. Intruders were consistently smaller and younger than residents, and no two mice were paired together more than once. Virgin 3–5mo female TAPP and WT mice were similarly tested for aggression at ZT23, but with juvenile (16–21d) group-housed female Swiss Webster intruders (which has been shown to produce an aggressive phenotype in females of some mouse strains[30]).

For each test, the total time that the resident mouse engaged in aggressive behaviors (including biting, wrestling, boxing, chasing, and tail rattling) was manually scored from the recorded video using Observer XT (Noldus) software by a trained investigator who was blinded to the genotype of the mouse and the time of day of the test. Behaviors of intruders were not scored.

## Brain preparation and sectioning

Following all tests, mice were deeply anesthetized with a ketamine/xylazine mixture and transcardially perfused using saline followed by 10% formalin. Brains were then removed and post-fixed for 2–4 h, and then transferred into a cryoprotectant sucrose solution until they sank. Brains were then frozen with dry ice and sectioned at 40 μm into three series using a microtome (American Optical 860).

## Fluorescent immunohistochemistry

For TAPP and WT mice, immunohistochemical analysis was performed on separate tissue series throughout the entire brain using antibodies against pTau or Aß. The pTau antibody was a mouse monoclonal (clone AT8, Invitrogen, cat# MN1020, lot# VD2974081, 1:5000) against partially purified human paired helical fragments Tau (PHF-Tau, hyperphosphorylated) which detects PHF-Tau (Ser202/Thr205) at a predicted molecular weight of approximately 79 kDa. The manufacturer validated using cell treatment that this antibody binds to the antigen. The Aß antibody was a mouse monoclonal (clone 6E10, Biolegend, cat# 803014, lot# B343930, 1:2000) against a 770 amino acid human APP. The manufacturer validated that this antibody recognizes the human Aβ peptide via Western blot, ELISA, and immunohistochemistry. The arginine vasopressin (AVP) antibody was a guinea pig polyclonal (BMA Biomedicals, cat# T5048, lot# A17901, 1:5000) against a synthetic peptide H-Cys-Tyr-Phe-Asn-Cys-Pro-Arg-Gly-NH₂ coupled to a carrier protein. This antibody was validated by the manufacturer in detecting (Arg8)-Vasopressin via ELISA. Secondary antibodies were donkey polyclonals raised against mouse serum, conjugated to Cy2 (Jackson Immunoresearch, cat# 715-225-150, lot# 146498, 1:500), or guinea pig serum, conjugated to Cy5 (Jackson Immunoresearch, cat# 706-175-148, lot# 144177, 1:500). These antibodies were validated by the manufacturer using immunoelectrophoresis and ELISA to bind to mouse IgG and guinea pig IgG, respectively. For cell counting, only labeled neurons with a clear, round nucleus were counted, and this was performed by a trained investigator who was blinded to the conditions of each mouse.

## RNAscope in situ hybridization combined with fluorescent immunohistochemistry

Following perfusion, as described above, brains were extracted and postfixed overnight in 10% formalin and then stored in 30% sucrose in 1X RNase free PBS at 4 °C, overnight and then sectioned using a freezing microtome (40 μm coronal sections into three series) under RNAse-free conditions. Following sectioning, tissue was stored at 4 °C in PBS containing the preservative sodium azide until processed for histology. Mid-brain sections from C-57 male or female mice were used for prodynorphin (pDyn) mRNA labeling by RNA scope in situ hybridization and cholera toxin subunit B (CTb) using immunohistochemistry. Brain sections from one of the series were mounted on Superfrost Plus slides under RNase-free conditions. Slides were air dried for at least 2 h at RT before storing at −80 °C until use. Slide-mounted sections were then baked in HybEZ oven for 20 min at 60 °C, and an RNAScope Multiplex Fluorescent Reagent Kit V2 (catalog# 323110, Advanced Cell Diagnostics) was applied over the tissue area. All sections were pretreated with hydrogen peroxide (catalog #322381, Advanced Cell Diagnostics) for 10 min at room temperature, and target retrieval (catalog #322000, Advanced Cell Diagnostics) was performed for 5 min by placing the slides in an oven (>99 °C). Sections were then dehydrated in 90% alcohol and air dried for 5 min, followed by drawing barrier around the sections using Immedge hydrophobic barrier pen (catalog# PN310018, Advanced Cell Diagnostics). The slides were washed in DEPC-PBS (diethylpyrocarbonate, Ambion) 2x2min at RT, followed by treating with protease reagent (Protease III, catalog #322381, Advanced Cell Diagnostics) for 30 min at 40 °C. After rinsing in sterile water, sections were incubated in RNA-scope probes for pDyn-C3 (RNAscope ProbeMm-pDyn-C3; catalog #318771-C3, Advanced Cell Diagnostics) for 2 h at 40 °C for hybridization. The slides were then washed with 1x Wash Buffer, provided in the kit, 2 × 2 min at RT and incubated in three amplification reagents at 40 °C (AMP1, AMP2 for 30 min each, and AMP3 for 15 min with washes after

each AMP step with 1x Wash buffer) followed by horseradish peroxidase-C3 (HRP-C3) amplification for 15 min at 40 °C and washed with 1× Wash buffer 2 × 2 min at RT. Sections were incubated in Opal fluorophores (Opal 690 [1:750], catalog # PN FP1497001KT, Akoya Biosciences) for 3 min at 40 °C to visualize mRNA. In the last step of the process, after washing, sections were subjected to HRP blocking for 15 min at 40 °C. Following in situ, the protocol for CTb immunofluorescence was performed. The sections with tissue were rinsed with 1x Wash buffer 2x2min at RT and incubated in a blocking solution of 3% horse serum in 1xPBT for 20 min at RT, then rinsed with 1xPBS 2 × 2min at RT and incubated with primary antibodies overnight. The CTb antibody was a mouse monoclonal (clone 23043, abcam, cat# ab62429, lot# GR3207107-16, 1:2000) against full length native CTb. This antibody was validated by the manufacturer in detecting CTb staining in the optic nerve following CTb intravitreal injection. Slides were washed with PBS 2 × 2min at RT and then incubated in secondary antibodies, donkey anti-mouse-TRITC (Invitrogen, cat# A24511, lot# 42-69-051914, 1:500), to visualize fluorescence. This secondary antibody was validated by the manufacturer as labeling mouse monoclonal antibodies against tubulin in HeLa cells. After a final PBS wash, slides were dried, and coverslipped with Fluoro-Gel mounting medium containing DAPI (Electron Microscopy Sciences), in order to stain nuclei. Cell counting was done similarly as described above, by a trained investigator blinded to the conditions of each mouse.

## Imaging

Sections labeled with immunohistochemistry and RNAscope in situ hybridization were scanned using a fluorescent microscope (Zeiss Axioscope) and imaging software (ZEN 3.2 blue).

## pTau quantification

Images of AT8-labeled LPB sections were analyzed using commercially available area calculator software (SketchandCalc, https://www.sketchandcalc.com/). Total area covered by pTau in the LPB was measured by outlining each cell and axon expressing AT8 labeling by a trained individual blind to the age and gender of each mouse. Two bilateral images through the middle of the LPB were used per mouse and an average was taken, in cases where bilateral images could not be used we generated unilateral images from consecutive slices. These averages were used to construct the regressions with Tb bathyphase and LMA acrophase for the same mouse.

## Statistics and sample size calculations

All data were imported to Prism 9 (Graphpad) and normality was assessed using the Shapiro-Wilk test to verify the appropriateness of the following statistical analyses. In cases where normality was violated, non-parametric approaches were undertaken and noted. TAPP and WT data were analyzed for hr-by-hr differences, area under the curve (AUC), Tb bathyphase and LMA acrophase differences. For hr-by-hr comparisons, two-way repeated measures ANOVA for main effects of TAPP vs WT, time, and interaction with Sidak's post hoc was used. For Tb bathyphase and LMA acrophase, unpaired two-tailed $T$ tests were used. For regression analysis, log-log regression with 95% confidence interval was used. For counting LPB$^{dyn}$ cells, one-way RM ANOVA with Sidak's post hoc tests was used. Aggression tests were analyzed using a two way RM ANOVA and Sidak's post hoc test, as in our prior published work[16], with the addition of the use of non-parametric Mann–Whitney tests to analyze non-normal data. For $pDyn$-Cre mice a pre-condition was recorded followed by neural injection of Cre-dependent caspase into the LPB and post condition recording after both two and four weeks. For hr-by-hr analyses, we used a two-way repeated measures ANOVA with Sidak's post hoc tests. For Tb bathyphase and LMA acrophase comparisons, we used a one-way repeated measures ANOVA with Sidak's post hoc tests. For area-under-the-curve (AUC) comparisons we used either a paired or unpaired $t$ test

where noted. All data are presented as mean ± standard error of the mean. For all tests, alpha was $p = 0.05$.

Sample size for each experiment was determined a priori with effect sizes calculated from means and standard deviations of early pilot experiments using G*Power software (v3). To determine sample size a priori, we used early pilot data on aggression from 7 to 9mo TAPP and WT males (means and standard deviations at ZT23), because in our previous work[16], aggression proved to be much more variable and require a larger sample size than LMA or Tb. We calculated an effect size of Cohen's $d = 0.767963414$, using the online Effect Size Calculator from Dr. Lee Becker from U Colorado, Colorado Springs (https://lbecker.uccs.edu). We then converted this to Cohen's $f = 0.383981707$, for use in G*Power software (v3), via the formula $d = 2f$. G*power software determined that, using settings of ANOVA: Repeated measures for within-between interaction, with 2 groups, 4 measurements, 85% power, and alpha = 0.05, a sample size of $n = 12$ was required to reliably detect a similar effect size. Our smallest sample size for these experiments in males ended up being well above this mark ($n = 18$, for 7–9mo TAPP male aggression), because we lost fewer males to attrition than expected (especially compared to females: 5% for males vs. 30% for females). Since it was apparent from our preliminary studies that TAPP and WT females less reliably show aggression compared to males (about 50% of females tested showed quantifiable levels of aggression), we instead determined sample size a priori for females using means and standard deviations from Tb bathyphase (our most reliable effect across all variables) from 3 to 5mo TAPP and WT females. For this data, we calculated an effect size (as described above) of Cohen's $d = 2.927542568$. We then converted this to Cohen's $f = 1.463771284$, as described above, for use in G*Power software. Using the settings of ANOVA: fixed effects, one-way, with 2 groups, 85% power, and alpha = 0.05, G*power software determined that a sample size of $n = 8$ was required to determine a similar effect. Accordingly, our lowest sample size among female groups was $n = 8$ (for 7–9mo TAPP females), with WT females having larger sample sizes due to us losing a high number of TAPP females to attrition (30%).

## Figure generation

Figures were created using Microsoft PowerPoint for Mac (version 16.73) or Adobe Photoshop (version 24.50). Graphs were made in Prism 9 (Graphpad) or Microsoft Excel for Mac (version 16.73). The schematic from Fig. 5d was created with Biorender.com. The schematic in Fig. 9 was made in Microsoft PowerPoint.

## Data availability

Source data are provided with this paper. All other data are available upon request. Source data are provided with this paper.

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

## Acknowledgements
We sincerely thank Clif Saper for materials, support, and guidance during the early stages of this study. We also thank Aaron Wilke, Taylor Haddock, Erik Gwaltney, Jacob Krul, Spence MacLellan, Kim Quintana, and Alexa Mejia for superb technical assistance. This work was supported by an Alzheimer's Association research fellowship (AARF 443613) to WDT, an NIH R03 grant (AG062883-02) to WDT, and NIH COBRE grants (3P20GM121310-05, 3P20GM121310-05S2, and 2P20GM121310-06 from NIGMS) wherein WDT is a Project Leader.

## Author contributions
A.E.W., P.G and W.D.T. designed experiments. A.E.W., P,G, M.M.R., Q.L.J., G.C.G., M.I.T., H.W.R. and W.D.T. ran experiments and analyzed data. A.E.W., P.G and W.D.T. wrote the manuscript. All authors read, edited, and contributed intellectually to the manuscript.

## Competing interests
The authors declare no competing interests.
