## [Peer Review File · Nature Communications]

A brainstem to circadian system circuit links Tau pathology to sundowning-related disturbances in Alzheimer's disease model miceReviewers' Comments:

Reviewer #1:

Remarks to the Author:

In this study by Warfield et al., the authors investigate the effects of tau pathology on a pathway from the lateral parabrachial nucleus to the subparaventricular zone and suprachiasmatic nucleus in wild-type and TAPP Alzheimer's model mice. They find that TAPP mice exhibit a disruption in body temperature and locomotor activity rhythms that coincides with the development of pTau in the LPB. Dynorphin neurons in the LPB project to the SCN and SPZ and a subset of these neurons coexpress pTau. LPB pTau is associated with increased behavioral aggression around subjective dawn. Finally, the authors find that genetically deleting LPB dynorphin neurons increases body temperature. The manuscript is well written and the statistics seem reasonable. Text on several of the figures is difficult to see and most of the immunohistochemistry images are difficult to interpret as presented. Overall, my biggest concern with the manuscript is that the causal link between pTau in pDyn LPB neurons, pDyn LPB to SPZ/SCN projections, and rhythm disruption (Tb, LMA, aggression) does not seem to be supported by the data as they are currently presented.

Specific concerns:

1) Actigraphy analysis

- a. The authors should elaborate on their choice of (different) phase marker for each of these rhythms.
- b. If the troughs/peaks calculated in Fig. 1 are delayed in TAPP mice, one would expect that the entire rhythm would be delayed regardless of the selected phase marker. This is not immediately apparent in Fig. 2, where it doesn't look like the trough of Tb or peak of LMA strongly differ between conditions. Changes in Tb/LMA "waveform" would be interesting but are different from a phase delay. The authors write that they observe "a clear rightward shifting of the curve" in Fig. 3, which I would agree is indeed a phase delay, but this isn't apparent in Fig. 2.
- c. How stable is the phase angle of entrainment in each mouse? The authors average across 7 days to obtain a single value per mouse. Even in wild-type mice, there is a variability of about 4 hours across mice which is quite broad. It would be more informative to plot the average acrophase on each day of recording (as in Escobar et al. 2020 Sci Rep, and many others) and perform statistics on each day of recording.
- d. Does phase angle of entrainment differ significantly between male and females? What about between 3-5mo and 7-9mo mice?
- e. The authors need to provide representative actograms for Tb and LMA activity of the WT and TAPP mice in each condition (age, sex).
- f. The authors should also provide representative actograms and (at least in the supplemental) include their Tb and LMA analysis for the DD condition instead of saying data not shown.
- g. In general the authors should be careful not to use "circadian effects" or similar when experiments are under entrained conditions.

2) LPB

- a. The authors should elaborate on their decision to focus on the LPB. In Supplemental Figure 2, pTau is strongly expressed throughout the brain of 7-9mo TAPP mice. The LPB does not seem to have greatly higher pTau expression than the surrounding brain regions.
- b. Representative IHC images for pTau in the LPB in Figs. 2, 3, 5, 6, etc. are difficult to interpret – it is not immediately apparent even with the provided arrows which condition has more or less pTau.
- c. The authors should quantify the amount of pTau in each of their conditions when possible (as in Supplemental Figure 1).
- d. The authors should provide quantification of their LPB to SPZ/SCN projections in Fig. 4 (n, variability between animals, etc.) in the text or in a figure subpanel.
- e. The sparse colocalization between pTau and LPB projections to the SPZ in Fig. 4d is concerning especially considering the central hypothesis of the manuscript. Are the cells that colocalize pDyn-positive? Do pTau+ LPB neurons project to the SCN? The authors state that this may be due to age-

related degeneration of LPB neurons – they observe that pDyn neurons degenerate in a subset of extremely old mice. The sparse colocalization may indeed be due to SPZ-projecting pTau+ LPB neurons degenerating, but there are still numerous pTau- LPB neurons that project to the SPZ.

3) Aggression

- a. The authors should justify their choice to use an unplanned comparison test in Fig. 6a in light of their two-way ANOVA results. They should also remove “did not rise to significance” from the text.
- b. Is there any aggressive behavior correlate in females? Sex is measured as a variable in the rest of the manuscript and the authors find prominent male-female differences. The authors do not necessarily need to do any additional experiments but should mention the lack of behavioral output for females as a limitation of the study in the discussion.
- c. Is there a positive correlation between pTau in pDyn LPB neurons and aggression? The authors show that the mouse with the highest level of aggression has a high level of pTau in the pDyn neurons, but pTau is upregulated throughout the LPB. If the claim is that the pTau in the pDyn LPB to SCN/SPZ circuit influences behavioral aggression, the authors should provide further evidence that pTau in pDyn LPB neurons (as opposed to pTau in the LPB in general) is correlated with aggression.
- d. It is unclear why the authors compared aggression in TAPP with and without pTau in Fig. 6c and in wild-type and TAPP without pTau in 6d. Comparisons in 3-5mo males and 7-9mo males should include all three variables.

4) LPB pDyn deletion

- a. In both males and females, LPB pDyn deletion appears to generally have a hyperthermic response instead of affecting the Tb rhythm per se. LPB neurons have been shown to be involved in thermosensation, and thermoresponsive pDyn LPB neurons project to the median preoptic nucleus to likely regulate temperature responses (e.g. Geerling et al. 2016).
- b. Summing total Tb and LMA during the dark and light phases and comparing them between conditions is generally not the best way to measure differences in rhythmicity. Also, the changes in phase angle of entrainment over time are difficult to interpret: e.g., Tb in males advancing in males at 2 weeks but delaying back to pre-deletion at 4 weeks, a temporal dissociation between phase angle of entrainment in females (phase advancing Tb but stable LMA).

5) Other

- a. Fig. 8 should probably be in the discussion.

Reviewer #2:

Remarks to the Author:

There is wide appreciation that sleep disruptions are common in AD patients and there has been the suggestion that disruption in the circadian timing system could lie at the heart of this symptoms. In this study, the authors examined the accumulation of Tau and beta-amyloid pathology and its association with circadian dysfunction in the TAPP (also called APPSwe-Tau) mouse model which express mutant human Tau and amyloid precursor protein mutations. They were particularly interested in whether sex differences in the development of such pathology influence the onset of circadian dysfunction and investigated which neural areas affected by such pathology also project to the circadian system.

The team started by looking for phase delays as a common feature of AD as well as to look for correlations with pTau throughout the brain. They looked at male and female mice at adult (3-5 mo) and middle age (7-9 mo). The authors report that females but not males exhibited a phase delay at the younger age. This is seen in both sexes at the middle age point.

The authors report a similar relationship with the expression of pTau in the lateral parabrachial nucleus (LPB) of the brainstem. I cannot see this myself. I would recommend that the author separate

out the histology from the physiological and behavioral analysis. How is the pTau quantified?

Fig. 2: Here the authors show the 1 hr bins and appropriately analyze the waveforms with two-way ANOVA. The phase delay is less obvious in this display. The reduction in amplitude of the rhythms is more obvious.

Fig. 3: The authors pullout the subset of TAPP males with pTau in LPB and look at waveform of these mice. In this subset of mice (n=5) the delays in LMA and Tb are quite clear. Note that the mutant mice are showing lower activity during the night. Note that the authors did not explore other brain regions of pTau.

Fig. 4: the authors injected a retrograde tracer (CTb) into both the SPZ and the SCN. They looked for co-expression of pDyn message with CTb in the LPB. I think that this looks pretty good. It would help to provide more details on how this data was analyzed.

In Fig. 5, they examine the co-expression of pTau and pDyn in the LPB. This analysis is done at 7-9 months and they see about 50% of the cells show coexpression. At older ages (12-13 month), they report a loss of pDyn neurons. They suggest but do not show that these cells are dying. There were not cell death assays. Not clear how co-expression was quantified. They are showing a reduction in the expression of pDyn.

Sundowning is at least partially characterized by agitation and aggression during this time in early night. Looking at the male TAPP mice it does look like an increase in aggression is observed at ZT 23 or just before lights ON. But the effect on behavior were stronger in the females so why looking at males? At ZT 23 in males at 7-9 months, we do see an increase in activity but not a significant one. So does this fit???

Based on these observations, the authors speculated that LPB dyn expressing neurons are responsible for changes in rhythms in activity, body temperature, and aggression. They sought to test this hypothesis by ablating these neurons in WT mice. They used genetically targeted ablations via a Cre-dependent caspase vector to kill LPBdyn cells in pDyn-IRES-cre mice and compared Tb and LMA rhythms. They need to better document the success of the lesions but did show example and cell counts. They found that the loss of LPBdyn increased Tb without effects on locomotion. The diurnal rhythms themselves did not appear to be altered. They did not appear to test aggression.

The authors argue for a model in which sundowning behavior is caused by pTau pathology in LPBdyn which lead to the death of these neurons. The loss of these cells leads to phase-delayed locomotor activity rhythms as well as an increase in aggression.

Strengths of this study include the development of a possible model for sundowning behavior which is characterized by wandering, agitation, and aggression in the late afternoon and early evening. The work is broad in ambition and uses several approaches to test their hypothesis. The methods used are generally clear.

Still I have a number of reservations about this work and do not feel that the data support the conclusions.

The writing itself is very dense and the flow of the logic is frequently lost.

The introduction needs to be trimmed with, for example, the paragraph about differences between AD models belonging in the discussion.

Throughout the ms the authors are making use of IHC to assess pTau, amyloid beta, cell death, pDyn expressing cells in LPB. They show a picture and sometime counts. But we need more information

about the quantification and in many cases the authors would be better served by showing a separate figure with the microscopic images and counts rather than putting them in the middle of figures showing other findings.

But fundamentally, I am just not convinced by the authors' arguments that they have the circuit underlying sundowning behavior which is characterized by wandering, agitation, and aggression in the late afternoon and early evening.

When I look at the activity and Tb waveforms, I do see an increase in activity in females at 7-9 months. The claimed phase delays are less clear and would need to be better documented. Perhaps using software like Clock-lab or other open source circadian analysis tools.

I am convinced that dramatic changes seen in the subset of male mice with pTau accumulation but here that activity is decreased. But how does this fit with the author's model that pTau accumulation would increase activity? And what the females TAPP mice who are showing the increase in activity at 7-9 months, should they not have increased pTau compared to males?

The move to aggression assays makes good sense. But here there are just looking at males and not the females that showed the increase in activity. They do see evidence for increased aggression in late night in the TAPP mice, but not in morning. This does not appear to fit with sundowning.

Genetically ablating these LPBdyn cells should increase activity during sleep time and increase aggression i.e. mimicking sundowning. Their data does not support this.

Therefore, I feel that much more work is needed to convince the reader that the authors have uncovered a circuit whose dysfunction underlies sundowning behavior.

Reviewer #3:

Remarks to the Author:

In this manuscript Warfield et al. examined the role of abeta and tau on circadian rhythm disruption in TAPP mouse model of Alzheimer's disease. The team reports phase delays in rhythms of body temperature and locomotor activity in TAPP mice compared to controls. They link these deficits to dynorphin-expressing LPB neurons that project to the circadian system and are particularly vulnerable to pTau pathology. This work has a number of major weaknesses that need to be addressed.

Major:

Results:

1. Fig 1: Show body temperatures across light-dark cycle. Same for locomotor activity. Absolute values are missing. That's figure 2. Present figure 2 before figure 1.
2. Fig 1J, if differences in locomotor activity are indeed due to neurodegeneration, then neurodegeneration differences in 7-9 month old males vs females should be measured and showed.
3. Fig 1: abeta pathology in addition to tau needs to be shown, since these mice have plaques and tangles. However, the ages assessed might be preceding significant plaque deposition. If that is the case, tau pathology precedes abeta pathology, which is inverse of what happens in Alzheimer's patients. That is problematic when trying to model human condition.
4. Fig. 2a,b,d,e,g,h,j,k missing x- and y-axes titles.
5. Fig. 2 bar graphs are too small, impossible to read.
6. Line 417: actual references are needed.
7. Fig 1-3: phosto tau needs to be quantified.
8. Fig 4: the rationale for using a light-activatable channel ChR2 is not clear. A more suitable marker would be GFP.
9. Fig 4d: quantification of ptau is missing.
10. Line 590: the following statement is unsubstantiated by data: We used 7-9mo males because this is the age and sex where there is consistent pTau pathology and limited impact of

neurodegeneration

Minor:

Methods: units for stereotaxic injection units are missing. AP: -0.05, ML: 175 +0.015, DV: -0.51), or over the LPB (AP: -0.54, ML: +0.14, DV: -0.30) Line 174-175

We sincerely thank the reviewers for their helpful suggestions and thoughtful critiques. We have incorporated their edits and feel that they have substantially improved our manuscript. We address each of their concerns below in **bold**.

Reviewer #1 (Remarks to the Author):

In this study by Warfield et al., the authors investigate the effects of tau pathology on a pathway from the lateral parabrachial nucleus to the subparaventricular zone and suprachiasmatic nucleus in wild-type and TAPP Alzheimer's model mice. They find that TAPP mice exhibit a disruption in body temperature and locomotor activity rhythms that coincides with the development of pTau in the LPB. Dynorphin neurons in the LPB project to the SCN and SPZ and a subset of these neurons coexpress pTau. LPB pTau is associated with increased behavioral aggression around subjective dawn. Finally, the authors find that genetically deleting LPB dynorphin neurons increases body temperature. The manuscript is well written and the statistics seem reasonable. Text on several of the figures is difficult to see and most of the immunohistochemistry images are difficult to interpret as presented. Overall, my biggest concern with the manuscript is that the causal link between pTau in pDyn LPB neurons, pDyn LPB to SPZ/SCN projections, and rhythm disruption (Tb, LMA, aggression) does not seem to be supported by the data as they are currently presented.

Specific concerns:

1) Actigraphy analysis

a. The authors should elaborate on their choice of (different) phase marker for each of these rhythms.

We initially focused on Tb bathyphase and LMA acrophase because they are phase markers of circadian entrainment that are commonly used in the human AD and sundowning literature (PMID: 11329390, 15879584, 25921596). However, we have also now added a table with Tb acrophase and LMA bathyphase to the supplemental figures (Table S1) and we see the same phase delays as reported for Tb bathyphase and LMA acrophase at the same age and sex.

b. If the troughs/peaks calculated in Fig. 1 are delayed in TAPP mice, one would expect that the entire rhythm would be delayed regardless of the selected phase marker. This is not immediately apparent in Fig. 2, where it doesn't look like the trough of Tb or peak of LMA strongly differ between conditions. Changes in Tb/LMA "waveform" would be interesting but are different from a phase delay. The authors write that they observe "a clear rightward shifting of the curve" in Fig. 3, which I would agree is indeed a phase delay, but this isn't apparent in Fig. 2.

This is an important point and we thank the reviewer for mentioning it. For calculating bathyphases and acrophases, we used ClockLab Analysis 6 which fits the data from each day to a sine function with a period of 24h and determines the trough and peak, respectively. We have added a more detailed description of these analyses in lines 176-183. Similar analyses for determining bathyphase and acrophase have been used in Tb and LMA data from human AD patients and such studies reliably report that these markers occur later compared to

healthy aged match controls – and this is consistently described as a phase delay (PMID: 11329390, 15879584, 25921596).

While we agree that not every point on the hr-by-hr graphs is shifted rightward on every graph in Figure 2, we would argue that this is evident in 3-5mo TAPP females for both Tb and LMA in that the increases around the light-to-dark transition are lagging behind the WT females. Also, for all TAPP mice that had significantly later bathyphases and acrophases, the decreasing values in the early light period occur at a later timepoint compared to their WT counterparts. We agree that the phase delay in 7-9mo TAPP mice is less obvious, however in these mice there is clear hyperthermia and hyperactivity in many of the time points of the dark phase and around the dark-to-light phase transition, which leads to a more prolonged descent into the trough (bathyphase) and an elongated ascent to the peak (acrophase) of both rhythms. This is visualized by the increased area under the curve (Figure S13) and in the hr-by-hr graphs in Figure 2. Again, in each of these cases the TAPP mice take longer to fall into their trough of body temperature and reach the peak of their activity later in the active phase.

It may be that some of the effects we are seeing are due to waveform changes, however, the bathyphase and acrophase analyses are the most relevant for our study in that they are the markers that have been extensively used in human AD patients, whereas we are less familiar with specific waveform analyses performed in such studies.

Also, the sampling rate for our biotelemetry system is every 5min whereas we binned this data into hr-by-hr averages so that it would be easier to detect individual timepoints where large differences occurred. It may be that binning this data into hours, or only sampling at 5min intervals, reduced the appearance of a rightward shift in some cases – yet the peaks and troughs of these rhythms would still be clear. We have added these caveats to lines 169-170 and 180-181.

c. How stable is the phase angle of entrainment in each mouse? The authors average across 7 days to obtain a single value per mouse. Even in wild-type mice, there is a variability of about 4 hours across mice which is quite broad. It would be more informative to plot the average acrophase on each day of recording (as in Escobar et al. 2020 Sci Rep, and many others) and perform statistics on each day of recording

The stability of the phase angle is indeed an interesting metric, so we have now calculated the standard deviation among the seven days that we had averaged for bathyphase and acrophase. This is in Figure S2 and described in lines 351-354. We did not perform statistics on individual days across groups because the actual day number within the consecutive seven days that we averaged is not necessarily meaningful since we are not counting days after a treatment or intervention of some kind. Furthermore, all mice had been previously acclimated to the circadian chambers for at least 5 days, as described in the Methods in lines 164-165. The average and standard deviation across the consecutive 7d analyzed in LD give a good look into the delay (average) and how stable either the acrophase or bathyphase is (standard deviation). We found no significant differences in the variability of these markers between TAPP and WT mice, except between 3-5mo TAPP mice who were first developing

LPB pTau (lines 351-354, Fig S5). We would argue that this increase in variability in mice who are in the earliest stages of LPB pTau development further supports or claim that the LPB[→]SCN/SPZ pathway is associated with circadian dysfunction. We have also included representative actograms to further show the stability and delay of our TAPP mice after LPB pTau development.

d. Does phase angle of entrainment differ significantly between male and females? What about between 3-5mo and 7-9mo mice?

These comparisons are shown in Figure S1 to which we have added comparisons between males and females. In short, the differences seen between sexes and between ages only appear when there is pTau accumulation in the LPB, with the exception of 7-9mo WT females being delayed compared to 3-5mo WT females (which offers an explanation for why 7-9mo TAPP females were not delayed compared to 7-9mo WT females in Fig 1).

e. The authors need to provide representative actograms for Tb and LMA activity of the WT and TAPP mice in each condition (age, sex).

We have added representative actograms for Tb and LMA for each of the possible conditions (Figures 3, 4, S3, and S4).

f. The authors should also provide representative actograms and (at least in the supplemental) include their Tb and LMA analysis for the DD condition instead of saying data not shown.

The added actograms (described above) include both LD (as shown in zoomed-in inset) and DD conditions (denoted by black vertical line to the left of each actogram). We have also added a table with the DD values (Table S2). We did not find any significant differences in free-running period length for either Tb or LMA in any of the conditions. This is described in lines 391-397.

g. In general the authors should be careful not to use “circadian effects” or similar when experiments are under entrained conditions.

This point is well taken since these terms can easily be overused. We have gone back through the manuscript and changed this term where appropriate to also include a specification regarding circadian entrainment. In some cases, we felt this to be an appropriate term since body temperature and locomotor activity are known to be circadian and we are measuring phase markers of circadian entrainment (acrophase and bathyphase) – although these are indeed under a light-dark cycle.

2) LPB

a. The authors should elaborate on their decision to focus on the LPB. In Supplemental Figure 2, pTau is strongly expressed throughout the brain of 7-9mo TAPP mice. The LPB does not seem to have greatly higher pTau expression than the surrounding brain regions.

There are multiple reasons that we focused on the LPB. We visually inspected several areas throughout the brain and consistently saw pTau AT8 labeling in the LPB of all mice in the

oldest age groups for both males and females (7-9mo). Additionally, our CTb retrograde tracing from the SCN and SPZ, followed by genetically targeted anterograde tracing from LPB^{dyn} neurons, revealed a novel circadian input pathway from this structure that has not been previously characterized. Additionally, the LPB region is one of the earliest areas to be affected by pTau in AD patients (PMID: 11515783), and we also found this to be the case in TAPP mice. This is further relevant because circadian dysfunction has been found to occur early in preclinical AD. Notably, in the 3-5mo male mice with pTau, the LPB was the only area that was consistently affected by pTau across all five, and all five experienced circadian disruption as shown in Figure 4. We also now show that other well-established circadian input structures (the midbrain raphe nuclei and the intergeniculate leaflet of the thalamus) lack detectable pTau in mice that have both circadian dysfunction and LPB pTau (Fig S11).

b. Representative IHC images for pTau in the LPB in Figs. 2, 3, 5, 6, etc. are difficult to interpret – it is not immediately apparent even with the provided arrows which condition has more or less pTau.

Figures 1 and 2 have been reorganized so that the bathyphase, acrophase, and hr-by-hr data are all in Figure 1 and the anatomy is in Figure 2 with larger images.

c. The authors should quantify the amount of pTau in each of their conditions when possible (as in Supplemental Figure 1).

We have now quantified the average area covered by pTau in the LPB per section and correlated area covered by pTau with both Tb bathyphase and LMA acrophase (in Figure 2). We did see a significant correlation between LPB pTau area and ZT23 agg in old males, but this was heavily dependent on the most aggressive mice and therefore did not meet regression assumptions. However, the graph from this regression is below and we will add it to the supplemental materials if the reviewers wish.

d. The authors should provide quantification of their LPB to SPZ/SCN projections in Fig. 4 (n, variability between animals, etc.) in the text or in a figure subpanel.

We have added a graph of the quantification of CTb colocalization with LPB^{dyn} neurons to Figure 4. The description of this quantification is in line 228-230.

e. The sparse colocalization between pTau and LPB projections to the SPZ in Fig. 4d is concerning especially considering the central hypothesis of the manuscript. Are the cells that colocalize pDyn-

positive? Do pTau+ LPB neurons project to the SCN? The authors state that this may be due to age-related degeneration of LPB neurons – they observe that pDyn neurons degenerate in a subset of extremely old mice. The sparse colocalization may indeed be due to SPZ-projecting pTau+ LPB neurons degenerating, but there are still numerous pTau- LPB neurons that project to the SPZ.

There are a few reasons why we could be seeing lower numbers of colocalization. The first being neurodegeneration, as described in lines 593-608. The second being that cells which have pTau may be affected by cytoskeletal and structural issues that limit the ability of the tracer to travel. Issues with axonal transport are common in Alzheimer's disease (PMID: 25278826). Finally, it is also possible that not all the damage done to the LPB by pTau is only done to the cells with accumulated pTau. It is likely that pTau is affecting the entire nucleus, given the high degree of interconnectivity within the LPB, not only the cells with high amounts of pTau would be negatively impacted.

3) Aggression

a. The authors should justify their choice to use an unplanned comparison test in Fig. 6a in light of their two-way ANOVA results. They should also remove “did not rise to significance” from the text.

We have removed “did not rise to significance”. The reason for our unplanned comparison (Mann Whitney test) was that the residuals in the two-way ANOVA were not normally distributed (from our Shapiro-Wilk test), so we sought to use a test that is not effected by the lack of normality of residuals.

b. Is there any aggressive behavior correlate in females? Sex is measured as a variable in the rest of the manuscript and the authors find prominent male-female differences. The authors do not necessarily need to do any additional experiments but should mention the lack of behavioral output for females as a limitation of the study in the discussion.

We have added female aggression data tested at ZT23 and added the graph to figure 7f. We focused on 3-5mo TAPP females (which we showed have pTau in the LPB at this age) and WT females and only tested at ZT23 given 1) the results seen in males at this timepoint when pTau is present on the LPB, 2) the fact that testing female aggression requires more subjects overall since only about 50% of female mice show aggression to begin with (as we found here and as reported in PMID: 28920934), and 3) to test female aggression you need juvenile female intruders which can only be used between the ages of 16-21d (PMID: 28920934).

c. Is there a positive correlation between pTau in pDyn LPB neurons and aggression? The authors show that the mouse with the highest level of aggression has a high level of pTau in the pDyn neurons, but pTau is upregulated throughout the LPB. If the claim is that the pTau in the pDyn LPB to SCN/SPZ circuit influences behavioral aggression, the authors should provide further evidence that pTau in pDyn LPB neurons (as opposed to pTau in the LPB in general) is correlated with aggression.

There is a positive correlation between the average area covered by pTau in the LPB and aggression at ZT23 in older mice. The graph of this regression is above in the response to Reviewer 1's comment 2c. However, the validity of this regression depends entirely on the most aggressive mice, so figuring out whether there truly is a positive correlation here will be a future aim of our lab. Importantly, mice have very different baseline levels of aggression and as such a linear association between amount of pTau and aggression may be difficult to begin with.

d. It is unclear why the authors compared aggression in TAPP with and without pTau in Fig. 6c and in wild-type and TAPP without pTau in 6d. Comparisons in 3-5mo males and 7-9mo males should include all three variables.

We compared the aggression in 3-5mo TAPP mice with pTau to those without as an extension of Figure 3 where these same mice showed later Tb bathyphases and LMA acrophases. The main idea being that the key difference between the 3-5mo male TAPP mice who do or do not show a delay and increased aggression is the presence or absence of pTau in the LPB. However, we have removed this graph. In all comparisons we were careful to include all variables.

4) LPB pDyn deletion

a. In both males and females, LPB pDyn deletion appears to generally have a hyperthermic response instead of affecting the Tb rhythm per se. LPB neurons have been shown to be involved in thermosensation, and thermoresponsive pDyn LPB neurons project to the median preoptic nucleus to likely regulate temperature responses (e.g. Geerling et al. 2016).

This is a very good point, we have added a few lines (967-970) discussing the likely role that the preoptic area plays.

b. Summing total Tb and LMA during the dark and light phases and comparing them between conditions is generally not the best way to measure differences in rhythmicity. Also, the changes in phase angle of entrainment over time are difficult to interpret: e.g., Tb in males advancing in males at 2 weeks but delaying back to pre-deletion at 4 weeks, a temporal dissociation between phase angle of entrainment in females (phase advancing Tb but stable LMA).

We have removed dark and light phase average graphs. We have also clarified what we think is going on with these phase markers in lines 793-798. The general hyperthermia we see after caspase ablation of LPB^{dyn} neurons is most likely masking any effects on phase change. We believe that the phase delay is due to the accumulation of pTau in the LPB while the general hyperthermia (as seen in the 7-9-month-old TAPP males and females) is likely due to silencing and eventual killing of the LPB^{dyn} cells as seen in aged TAPP mice in figure 4.

5) Other

a. Fig. 8 should probably be in the discussion.

We agree and have now moved Figure 8 into the discussion.

Reviewer #2 (Remarks to the Author):

There is wide appreciation that sleep disruptions are common in AD patients and there has been the suggestion that disruption in the circadian timing system could lie at the heart of this symptoms. In this study, the authors examined the accumulation of Tau and beta-amyloid pathology and its association with circadian dysfunction in the TAPP (also called APPSwe-Tau) mouse model which express mutant human Tau and amyloid precursor protein mutations. They were particularly interested in whether sex differences in the development of such pathology influence the onset of circadian dysfunction and investigated which neural areas affected by such pathology also project to the circadian system.

The team started by looking for phase delays as a common feature of AD as well as to look for correlations with pTau throughout the brain. They looked at male and female mice at adult (3-5 mo) and middle age (7-9 mo). The authors report that females but not males exhibited a phase delay at the younger age. This is seen in both sexes at the middle age point.

The authors report a similar relationship with the expression of pTau in the lateral parabrachial nucleus (LPB) of the brainstem. I cannot see this myself. I would recommend that the author separate out the histology from the physiological and behavioral analysis. How is the pTau quantified?

We have now added pTau quantification (Figure 2) and a description of this method in lines 273-280. We have also moved these images to this figure and increased their size.

Fig. 2: Here the authors show the 1 hr bins and appropriately analyze the waveforms with two-way ANOVA. The phase delay is less obvious in this display. The reduction in amplitude of the rhythms is more obvious.

This is similar to Reviewer 1's comment (1b), and is addressed above. We would argue that these graphs do show a phase delay for both LMA and Tb in the 3-5 month-old females in Figure 2 and for the 3-5 month-old males in figure 3 because, in all cases the it takes later to reach the trough of body temperature and the peak of locomotor activity. The delay is less obvious in old mice where general hyperthermia seems to be the more dominant phenotype. However, it is important to note that since the 7-9 month-old TAPP males and females have increased LMA and Tb at the hours around the active to inactive phase transition, it takes them longer to reach the trough of their body temperature rhythm and the peak of their activity rhythm.

Fig. 3: The authors pullout the subset of TAPP males with pTau in LPB and look at waveform of these mice. In this subset of mice (n=5) the delays in LMA and Tb are quite clear. Note that the mutant mice are showing lower activity during the night. Note that the authors did not explore other brain regions of pTau.

The 3-5mo TAPP males with LPB pTau do show lower, although not significantly lower, activity at the active to inactive transition. This is an unanswered question and a topic of future research in our lab. It is also important to note however, that these mice are at a very early stage of pTau accumulation – one that may precede where patients with some of the more “hallmark” symptoms of sundowning are, including increased wandering. It should also be noted that LMA rhythms peak in the very early active phase and so it is very possible that, in the case of a phase delay, mice never really have the chance to reach this peak because by the time their LMA starts to ramp up, it is already beyond the point where that peak often happens.

Fig. 4: the authors injected a retrograde tracer (CTb) into both the SPZ and the SCN. They looked for co-expression of pDyn message with CTb in the LPB. I think that this looks pretty good. It would help to provide more details on how this data was analyzed.

We added a description of this to lines 228-230. The data was analyzed via manual cell counting by trained individuals who were blind to the sex and age of the mice. Only cells with a clear, round nucleus were counted.

In Fig. 5, they examine the co-expression of pTau and pDyn in the LPB. This analysis is done at 7-9 months, and they see about 50% of the cells show coexpression. At older ages (12-13 month), they report a loss of pDyn neurons. They suggest but do not show that these cells are dying. There were not cell death assays. Not clear how co-expression was quantified. They are showing a reduction in the expression of pDyn.

Co-expression was quantified similar to how CTb/pdyn neurons were counted (via manual cell counting by trained individuals who were blind to the sex and age of the mice. Only cells with a clear, round nucleus were counted). We are indeed showing a reduction in the expression of pDyn and state that this is likely due neurodegeneration. In the case of the representative 12-13mo female TAPP mouse shown in Fig6 and Supp Fig 8, we see a “hole” or lesion in the tissue which is indicative of cell death. We also show a DAPi image in this case (Supp Fig 8) to further show less overall cells in the LPB region.

Sundowning is at least partially characterized by agitation and aggression during this time in early night. Looking at the male TAPP mice it does look like an increase in aggression is observed at ZT 23 or just before lights ON. But the effect on behavior were stronger in the females so why looking at males? At ZT 23 in males at 7-9 months, we do see an increase in activity but not a significant one. So does this fit???

We looked initially at males because aggression is better understood and more commonly studied in males and our initial experiments in female mice with male juvenile intruders did not produce quantifiable levels of aggression in females. However, we have added female aggression measured using the same resident intruder paradigm, though with female juvenile mice which was recently established to produce more readily quantifiable levels of aggression in some strains (PMID: 28920934). We focused on 3-5mo because TAPP females reliably show LPB pTau at this age and ZT23 because this is the timepoint that showed the

greatest increase in aggression in TAPP males at ages when they reliably express LPB pTau. We now show that these female TAPP mice also show increased aggression at ZT23.

The trend towards increased LMA at ZT23 and the increased aggression at ZT23 are different measures. Both measurements are complex behavioral outputs which do not necessarily track together. Notably, our prior research and that of others has shown that the rhythms of LMA and aggression propensity are regulated separately by the SCN through different SPZ pathways (ventral SPZ versus dorsal SPZ, respectively) (PMID: 11425913, 29632359, 23706187).

Based on these observations, the authors speculated that LPB dyn expressing neurons are responsible for changes in rhythms in activity, body temperature, and aggression. They sought to test this hypothesis by ablating these neurons in WT mice. They used genetically targeted ablations via a Cre-dependent caspase vector to kill LPBdyn cells in pDyn-IRES-cre mice and compared Tb and LMA rhythms. They need to better document the success of the lesions but did show example and cell counts. They found that the loss of LPBdyn increased Tb without effects on locomotion. The diurnal rhythms themselves did not appear to be altered. They did not appear to test aggression.

We have now added a graph comparing average number of pDyn neurons in WT mice to number remaining in caspase ablated mice showing a 91% reduction in pDyn neurons. We have also added a graph correlating the number of remaining LPB^{dyn} cells and the change in total area under the curve (a measure of the observed hyperthermia) which shows that the more successful ablations produced more hyperthermia. We did not test aggression in these mice because the males used here were not sexually experienced, a requirement for the resident intruder paradigm in males.

The authors argue for a model in which sundowning behavior is caused by pTau pathology in LPBdyn which lead to the death of these neurons. The loss of these cells leads to phase-delayed locomotor activity rhythms as well as an increase in aggression.

Strengths of this study include the development of a possible model for sundowning behavior which is characterized by wandering, agitation, and aggression in the late afternoon and early evening. The work is broad in ambition and uses several approaches to test their hypothesis. The methods used are generally clear.

Still I have a number of reservations about this work and do not feel that the data support the conclusions.

The writing itself is very dense and the flow of the logic is frequently lost.

We have made edits to improve the flow of our writing, especially to the writing that corresponds to the first two images. We reorganized figures 1 and 2 to have recording data first and then anatomy second. This allowed us to take a clearer approach.

The introduction needs to be trimmed with, for example, the paragraph about differences between AD models belonging in the discussion.

We have trimmed the introduction and moved the relevant parts of this section to the discussion.

Throughout the ms the authors are making use of IHC to assess pTau, amyloid beta, cell death, pDyn expressing cells in LPB. They show a picture and sometime counts. But we need more information about the quantification and in many cases the authors would be better served by showing a separate figure with the microscopic images and counts rather than putting them in the middle of figures showing other findings.

We have added a quantification of average area covered by pTau in the LPB and correlated this with both Tb bathyphase and LMA acrophase which is now in Figure 2. We also changed the format of Figure 1 and 2 with the anatomy images moved to Figure 2. Figure 4, 5, and 6 have also been altered to make the images larger.

But fundamentally, I am just not convinced by the authors' arguments that they have the circuit underlying sundowning behavior which is characterized by wandering, agitation, and aggression in the late afternoon and early evening. When I look at the activity and Tb waveforms, I do see an increase in activity in females at 7-9 months. The claimed phase delays are less clear and would need to be better documented. Perhaps using software like Clock-lab or other open source circadian analysis tools.

We used ClockLab Analysis 6 in the analysis of all phase markers. Clocklab fits the data from each day to a sine function with a period of 24h, and calculates trough (bathyphase) and peak (acrophase) for each day based on local time. We convert this value into hours from the lights-on (for bathyphase) or lights-off (for acrophase) transitions (since the difference between 24:00 and 0:00 is one hour, not 24) and average these values over 7d. We have added more details about these analyses in line 176-180 and now provide representative actograms (Fig3,4 and Supp Fig 3,4).

I am convinced that dramatic changes seen in the subset of male mice with pTau accumulation but here that activity is decreased. But how does this fit with the author's model that pTau accumulation would increase activity? And what the females TAPP mice who are showing the increase in activity at 7-9 months, should they not have increased pTau compared to males? The move to aggression assays makes good sense. But here there are just looking at males and not the females that showed the increase in activity. They do see evidence for increased aggression in late night in the TAPP mice, but not in morning. This does not appear to fit with sundowning.

We agree that it is curious that the activity its decreased in the subset of 3-5mo TAPP males with LPB pTau, albeit non-significantly, at the hours around the active-inactive phase transition. One potential reason could be that there an interaction between age and LPB pTau that contributes to increased LMA at particular times, whereas when pTau is present at earlier ages it causes mainly a shift (delay) in the rhythm (as bathyphases and acrophases are significantly later in this subset of mice). Additionally, and as mentioned above in the section on Fig 3, a phase delay could itself limit LMA in these mice especially at the inactive-active phase transition. We have added a short discussion of this possibility to lines 411-414.

3-5 month females do have increased LPB pTau compared to 3-5mo males w/LPB pTau and this can now be seen in our pTau quantifications in Figure 2. The reason pTau appears less in 7-9mo TAPP females compared to 3-5mo TAPP females is likely due to neurodegeneration, which we see strikingly in some female mice when we let them age to 13mo (Figure 6). If LPB cells are dying in 7-9mo TAPP females, who develop such LPB pTau much earlier than TAPP males, then there will be less intracellular pTau to stain for. We now have aggression data from 3-5mo TAPP and WT females at ZT23 and their aggression is increased just like in males that similarly have pTau.

The majority of research on sundowning suggests that the primary symptoms of this clinical phenomenon are “psychomotor agitation” including aggression and wandering that most prominently occur during a 4h window from roughly 4-8pm local time (this was a major focus of a recent review from our lab, PMID: 3301330, cited in this manuscript). This 4h window includes the late afternoon and early evening for diurnal humans (the transition from their active phase to their rest phase). For mice, ZT23 is late in the active phase (the subjective “late afternoon” for a nocturnal species). ZT0 and ZT1 are early in the active phase for mice (the subjective “early evening” for a nocturnal species). We see changes in aggression, LMA, or Tb at one of these timepoints in all TAPP mice that have LPB pTau (ZT23, ZT0, or ZT1). Aggression is significantly increased in 3-5mo TAPP females and 7-9mo TAPP males at ZT23 (which all reliably show LPB pTau at these ages), and in the subset of 3-5mo TAPP males that developed LPB pTau early at ZT1. LMA is significantly increased in 7-9mo TAPP females at ZT23, in 7-9mo TAPP males at ZT0, and there is a trend toward a similar increase in 3-5mo TAPP females at ZT23. Tb is significantly increased in 3-5mo TAPP females at ZT23, in 7-9mo TAPP females at ZT23, ZT0, and ZT1, in 7-9mo TAPP males at ZT23, ZT0, and ZT1, and in the subset of 3-5mo TAPP males that already developed LPB pTau at ZT23 and ZT0.

The fact that differences in Tb, LMA, and aggression in TAPP mice occur at slightly different times within the same mice as well as across age and sex groups is unsurprising because it is well-established that the SCN regulates these different rhythms through separate pathways in the SPZ, which project to separate downstream structures. Thus, slight individual variations in the amount of LPB pTau may account for which part of the SPZ (or SCN) is most affected and thus which rhythms are most impacted in a particular mouse and at precisely which timepoint around this active-to-rest phase transition. But our data strongly support that LPB pTau increases these variables within this 3h window (ZT23-ZT1), which is similar to the 4h window reported in AD patients that exhibit sundowning.

Genetically ablating these LPB^{dyn} cells should increase activity during sleep time and increase aggression i.e. mimicking sundowning. Their data does not support this.

We are not arguing that ablating LPB^{dyn} cells is a model for sundowning, but that it recapitulates part of the phenotype of our older TAPP mice, i.e. hyperthermia. It should also be stated that patients with AD have been shown to have elevated body temperature throughout the 24h day (general hyperthermia) compared to both young and old healthy control subjects (PMID: 15879584, reference 6 in manuscript). In this way, the increased body temperature reflects a later stage of LPB^{dyn} cell death than what we see in TAPP mice

which likely have dysfunctional (though not completely ablated) LPB neurons due to intracellular pTau.

Therefore, I feel that much more work is needed to convince the reader that the authors have uncovered a circuit whose dysfunction underlies sundowning behavior.

While we appreciate the reviewer's concern, we are not claiming that the circuit we characterize here accounts for all aspects of sundowning which, like Alzheimer's disease, is a uniquely human condition. However, transgenic mice that incorporate human mutations and develop AD-related pathology in the appropriate areas (like TAPP mice) serve as useful models of what such pathology does to mouse behavior and physiology. Of course, human behavior and physiology are much more complex than in mice, but understanding such effects in mouse models can point to changes in underlying behavioral and physiological states that are a major component of similar time-dependent changes in humans. Our central claim is that the temporal disturbances we see in TAPP mice at the active-to-rest phase transition are highly relevant for those seen in AD patients around their active-to-rest phase transition, and that further research into the role of pTau in the LPB in human AD patients (which has been extensively reported but never in the context of circadian dysfunction) is warranted and could provide a starting point for better understanding sundowning.

Reviewer #3 (Remarks to the Author):

In this manuscript Warfield et al. examined the role of abeta and tau on circadian rhythm disruption in TAPP mouse model of Alzheimer's disease. The team reports phase delays in rhythms of body temperature and locomotor activity in TAPP mice compared to controls. They link these deficits to dynorphin-expressing LPB neurons that project to the circadian system and are particularly vulnerable to pTau pathology. This work has a number of major weaknesses that need to be addressed.

Major:

Results:

1. Fig 1: Show body temperatures across light-dark cycle. Same for locomotor activity. Absolute values are missing. That's figure 2. Present figure 2 before figure 1.

We have combined the phase marker and hr-by-hr data into Figure 1 and have now moved the anatomical images to Figure 2 and present it along with our quantifications of pTau.

2. Fig 1J, if differences in locomotor activity are indeed due to neurodegeneration, then neurodegeneration differences in 7-9 month old males vs females should be measured and showed.

We believe that the differences in phase are due to the appearance of pTau in the LPB→SCN/SPZ circuit not necessarily the neurodegeneration. Such intracellular pTau in LPB neurons likely alters the function before they start to degenerate, and there is evidence for behavioral and physiological disruptions occurring prior to neurodegeneration in similar models (PMID: 18661556). However, such neurodegeneration ultimately follows the accumulation of pTau in the LPB.

3. Fig 1: abeta pathology in addition to tau needs to be shown, since these mice have plaques and tangles. However, the ages assessed might be preceding significant plaque deposition. If that is the case, tau pathology precedes abeta pathology, which is inverse of what happens in Alzheimer's patients. That is problematic when trying to model human condition.

We have added more a-beta images to the supplementary figures (Figures S9 and S10). We are not making the claim that pTau precedes a-beta in TAPP mice and in fact, we show a-beta at similar ages as the appearance of pTau in TAPP mice. However, we found a-beta pathology mainly in cortical and hippocampal regions, and we did not consistently see it in areas that also provide direct input to the SCN and SPZ based on our CTb labeling and previous work (PMID: 32879310).

4. Fig. 2a,b,d,e,g,h,j,k missing x- and y-axis titles.

We have added x- and y-axis titles where appropriate.

5. Fig. 2 bar graphs are too small, impossible to read.

Bar graphs have been removed.

6. Line 417: actual references are needed.

We have added these references.

7. Fig 1-3: phostotau needs to be quantified.

pTau has now been quantified and put into Figure 2.

8. Fig 4: the rationale for using a light-activatable channel ChR2 is not clear. A more suitable marker would be GFP.

ChR2 is an effective anterograde tracer, and it is engineered to transduce axon terminals well since these are often what is stimulated in optogenetic experiments. ChR2 has also been similarly used as a tracer in previous studies by other groups (PMID: 28039375, 30893297).

9. Fig 4d: quantification of ptau is missing.

Quantification of pTau has been added to Figure 2 and described in lines 273-280.

10. Line 590: the following statement is unsubstantiated by data: We used 7-9mo males because this is the age and sex where there is consistent pTau pathology and limited impact of neurodegeneration

This line has been removed.

Minor:

Methods: units for stereotaxic injection units are missing. AP: -0.05, ML: 175 +0.015, DV: -0.51), or over the LPB (AP: -0.54, ML: +0.14, DV: -0.30) Line 174-175.

We have added units to our injection coordinates (lines 148-149).

Reviewers' Comments:

Reviewer #1:

Remarks to the Author:

In the revised manuscript by Warfield et al., the authors address several concerns made by the reviewers. They have added further analysis in the supplemental material and changed several figures. However, the causal link between pTau in pDyn LPB neurons, pDyn LPB to SPZ/SCN projections, and rhythm disruption (Tb, LMA, aggression) is still not supported by the data as they are currently presented.

Major concerns:

1) The representative IHC images are still impossible to interpret. For instance, Fig. 2b, c, d supposedly have more pTau than 2b', 2c', 2d', but the images are almost indistinguishable – there does not appear to be any major difference in the green fluorescence in the areas immediately surrounding the white area indicating the LPB. The borders of the LPB should be marked on the images. Differences in the levels of pTau in the LPB is a critical part of this manuscript so it should be immediately apparent from the representative images that there are differences. At minimum the authors should include as a supplemental figure an example IHC image with the AT8-positive cells and axons outlined as discussed in the pTau quantification section of the methods.

2) Locomotor activity analysis remains confusing and difficult to interpret. The calculation of peaks and troughs for LMA and Tb using a sine fit on each day, averaged across each mouse over 7 days, compared across genotypes, can be misleading. I strongly recommend the mean and error (averaged across mice) be plotted on each day of the 7 days of LD recording for each genotype. Are the peaks and troughs delayed on each day, or only some days?

3) The conclusion that pDyn LPB neurons that project to the SCN/SPZ preferentially express pTau which explains the actigraphy differences between mouse genotypes is not supported by the data. For the authors' model to be supported, it is essential that they show a (relative) upregulation of pTau in SPZ/SCN projecting LPB pDyn neurons. The authors show that neurodegeneration is enhanced in LPB pDyn cells in aged TAPP mice, but LPB pDyn ablation does not recapitulate the actigraphy differences seen in the TAPP mice.

Minor concerns:

1) Most if not all of the figures need to be presented more clearly, e.g. Fig. 1 LMA and Tb plots, the axes are difficult to read, the figures are poorly aligned, plots are squished or stretched making the fonts look strange, etc.

2) Several sentences in the results section need to be moved to the discussion, such as lines 384-385 "monitoring these rhythms could provide clinicians with a tool for early detection of pTau pathology."

Reviewer #2:

Remarks to the Author:

The authors have substantially revised the manuscript and have dealt with my prior concerns.

In particular, my concerns about the image quality as well as the analysis of the rhythmic data have been addressed.

I believe that the revised work will be an important addition to the literature.

Reviewer #3:

Remarks to the Author:

The authors adequately addressed my concerns.

We thank Reviewer 1 for their suggestions regarding the IHC images and pTau quantification, and presentation of other Figures, and believe these changes further improved the manuscript. We appreciate their concerns and have addressed them below in **bold**.

Reviewer #1 (Remarks to the Author):

In the revised manuscript by Warfield et al., the authors address several concerns made by the reviewers. They have added further analysis in the supplemental material and changed several figures. However, the causal link between pTau in pDyn LPB neurons, pDyn LPB to SPZ/SCN projections, and rhythm disruption (Tb, LMA, aggression) is still not supported by the data as they are currently presented.

Major concerns:

1) The representative IHC images are still impossible to interpret. For instance, Fig. 2b, c, d supposedly have more pTau than 2b', 2c', 2d', but the images are almost indistinguishable – there does not appear to be any major difference in the green fluorescence in the areas immediately surrounding the white area indicating the LPB. The borders of the LPB should be marked on the images. Differences in the levels of pTau in the LPB is a critical part of this manuscript so it should be immediately apparent from the representative images that there are differences. At minimum the authors should include as a supplemental figure an example IHC image with the AT8-positive cells and axons outlined as discussed in the pTau quantification section of the methods.

We have edited Figure 2 so that the representative examples appear larger and have also outlined the boundaries of the LPB in red. In Figure S3 we now also include images of these same sections with pTau pathology outlined in our area quantification software.

2) Locomotor activity analysis remains confusing and difficult to interpret. The calculation of peaks and troughs for LMA and Tb using a sine fit on each day, averaged across each mouse over 7 days, compared across genotypes, can be misleading. I strongly recommend the mean and error (averaged across mice) be plotted on each day of the 7 days of LD recording for each genotype. Are the peaks and troughs delayed on each day, or only some days?

Please see next page for day-by-day graphs, statistical analyses, and response.

Day by Day Tb Bathyphase

TAPP w/pTau vs. TAPP w/o pTau:

TAPP w/pTau vs. WT:

A: *
B: **
C: ***
D: ****

Day by Day LMA Acrophase

TAPP w/pTau vs. TAPP w/o pTau:

TAPP w/pTau vs. WT:

A: *
B: **
C: ***
D: ****

As can be seen in the above graphs, TAPP mice (red lines) at each of the conditions where LPB pTau is seen show later (delayed) Tb bathyphase and LMA acrophase for the majority of the seven days recorded compared to either age- and sex-matched WT controls or age- and sex-matched TAPP mice without LPB pTau. We analyzed these graphs by comparing each day individually between conditions using either unpaired t-tests or non-parametric Mann-Whitney tests when the assumption of normality in model errors was violated. We did not use a two-way Anova because of the violation of normality.

It is important to note however, that this type of an analysis assumes that each particular day (i.e., “day 1, “day 2”, etc.) is meaningful compared to each other day, when in our current context they are not. We

allowed mice to acclimate for at least 5 days in their chambers and then began recording and this first full day of recording was deemed "day 1", but this "day 1" was arbitrarily assigned in the sense that it is not day 1 after an intervention. In contrast, the paper cited by Reviewer 1 in their first review (Escobar et al. Sci Rep, 2020), did have an intervention. In this study, the investigators subjected the mice to a six-hour phase advance and then measured acrophase values on the following days in control mice and mice that were given chocolate. In this case, "day 1" would be meaningful because it is the first day after an intervention (the phase advance). However, in our case we are simply recording undisturbed behavior, following an extended acclimation period, for seven consecutive days in each condition. Thus, these days are not meaningfully numbered other than being consecutive. Despite this, we now include the day-by-day analysis here and the same pattern of a consistent phase delay in every group where pTau is present in the LPB can be seen in this format. For the above reasons, in the last submission we had compared the standard deviations within the seven consecutive days between conditions because this more appropriately answers Reviewer 1's original (and valuable) question of how stable the bathyphases and acrophases and bathyphases are, and we reported this in Fig S2. However, we will also include this current day-by-day analysis in a supplementary figure if the Editor wishes.

3) The conclusion that pDyn LPB neurons that project to the SCN/SPZ preferentially express pTau which explains the actigraphy differences between mouse genotypes is not supported by the data. For the authors' model to be supported, it is essential that they show a (relative) upregulation of pTau in SPZ/SCN projecting LPB pDyn neurons. The authors show that neurodegeneration is enhanced in LPB pDyn cells in aged TAPP mice, but LPB pDyn ablation does not recapitulate the actigraphy differences seen in the TAPP mice.

In Figure 5a-c we show that about 50% of the LPB^{→SCN/SPZ} neurons express pDyn. We are not claiming that *only* the pDyn-expressing LPB^{→SCN/SPZ} neurons contribute to all aspects of the TAPP phenotype. Instead, we show in Figure 8 and explain in lines 817-835, that LPB^{dyn} ablations recapitulate the general hyperthermia seen in older TAPP mice, but not the phase delay. As seen in Figure 1c,g and Figure S13, this general hyperthermia is seen in older TAPP mice at ages consistent with neurodegeneration of LPB^{dyn} cells. We believe that the presence of pTau and its induced dysfunction (prior to neurodegeneration) in LPB^{dyn} neurons, as well as most likely in another partially overlapping LPB population (which accounts for the 50% of CTb cells in the LPB which were not pDyn+), is responsible for the phase delay and that the subsequent neurodegeneration is largely to blame for the hyperthermic phenotype seen in older TAPP mice. Nevertheless, our current manuscript strongly implicates pTau in the LPB in the phase delays (Figures 1, 2, and 4) and increased Tb, LMA, and aggression around the active-to-rest phase transition (Figures 1, 4, and 7). We also already show the presence of pTau in LPB^{→SCN/SPZ} neurons in Figure 5f and in about 50% of LPB^{dyn} neurons in Figure 6e. We agree that it would be ideal if we could have quantified pTau, CTb, and pDyn neurons in TAPP mice. But as we discuss in lines 616-619 of this resubmission, and in our previous Response to Reviewers, it is difficult to quantify CTb labelling in these TAPP mice due to the fact that 1) LPB neurons, including LPB^{dyn} neurons, are subject to neurodegeneration in TAPP mice (as we show in Figure 6), and 2) the existence of pTau (a cytoskeletal protein) in LPB^{→SCN/SPZ} neurons potentially interferes with the ability of CTb to travel retrogradely from the SCN/SPZ to the LPB. Future studies in our lab will aim to more thoroughly interrogate the role of pTau *only* in LPB^{→SCN/SPZ} neurons versus the LPB^{dyn} population and other LPB subpopulations, but given the amount of data we already show (8 data figures, 14 supplementary data figures, and 2 supplementary data tables) we believe this is outside the scope of the current manuscript.

Minor concerns:

1) Most if not all of the figures need to be presented more clearly, e.g. Fig. 1 LMA and Tb plots, the axes are difficult to read, the figures are poorly aligned, plots are squished or stretched making the fonts look strange, etc.

We have made the suggested changes to Figure 1 and improved the resolution of the text and the alignment of the graphs. We also made similar changes to Figures 4, 5, and 8.

2) Several sentences in the results section need to be moved to the discussion, such as lines 384-385 "monitoring these rhythms could provide clinicians with a tool for early detection of pTau pathology."

We have moved these lines to the discussion.

Reviewers' Comments:

Reviewer #1:

Remarks to the Author:

The authors have satisfactorily addressed my concerns. I thank them for addressing my concerns for a second time! I endorse the manuscript for publication.